# Serine Protease HtrA2/Omi Deficiency Impairs Mitochondrial Homeostasis and Promotes Hepatic Fibrogenesis via Activation of Hepatic Stellate Cells

**DOI:** 10.3390/cells8101119

**Published:** 2019-09-20

**Authors:** Wonhee Hur, Byung Yoon Kang, Sung Min Kim, Gil Won Lee, Jung-Hee Kim, Min-Kyung Nam, Hyangshuk Rhim, Seung Kew Yoon

**Affiliations:** 1Catholic University Liver Research Center, College of Medicine, The Catholic University of Korea, Seoul 06591, Korea; wendyhur@catholic.ac.kr (W.H.); kby2132@catholic.ac.kr (B.Y.K.); ring704@nate.com (S.M.K.); lgw0429@catholic.ac.kr (G.W.L.); kim.jh@catholic.ac.kr (J.-H.K.); 2Department of Biomedicine and Health Sciences, College of Medicine, The Catholic University of Korea, Seoul 06591, Korea; wangmouse@catholic.ac.kr (M.-K.N.); hrhim@catholic.ac.kr (H.R.)

**Keywords:** mitochondrial function, hepatic fibrogenesis, HtrA2/Omi, reactive oxygen species stress, mitochondrial homeostasis

## Abstract

The loss of mitochondrial function impairs intracellular energy production and potentially results in chronic liver disease. Increasing evidence suggests that mitochondrial dysfunction in hepatocytes contributes to the activation of hepatic stellate cells (HSCs), thereby resulting in hepatic fibrogenesis. High-temperature requirement protein A2 (HtrA2/Omi), a mitochondrial serine protease with various functions, is responsible for quality control in mitochondrial homeostasis. However, little information is available regarding its role in mitochondrial damage during the development of liver fibrosis. This study examined whether HtrA2/Omi regulates mitochondrial homeostasis in hepatocyte during the development of hepatic fibrogenesis. In this study, we demonstrated that HtrA2/Omi expression considerably decreased in liver tissues from the CCl_4_-induced liver fibrotic mice model and from patients with liver cirrhosis. Knockdown of HtrA2/Omi in hepatocytes induced the accumulation of damaged mitochondria and provoked mitochondrial reactive oxygen species (mtROS) stress. We further show that the damaged mtDNA isolated from HtrA2/Omi-deficient hepatocytes as a form of damage-associated molecular patterns can induce HSCs activation. Moreover, we found that motor neuron degeneration 2-mutant mice harboring the missense mutation Ser276Cys in the protease domain of HtrA2/Omi displayed altered mitochondrial morphology and function, which increased oxidative stress and promoted liver fibrosis. Conversely, the overexpression of HtrA2/Omi via hydrodynamics-based gene transfer led to the antifibrotic effects in CCl_4_-induced liver fibrosis mice model through decreasing collagen accumulation and enhancing anti-oxidative activity by modulating mitochondrial homeostasis in the liver. These results suggest that suppressing HtrA2/Omi expression promotes hepatic fibrogenesis via modulating mtROS generation, and these novel mechanistic insights involving the regulation of mitochondrial homeostasis by HtrA2/Omi may be of importance for developing new therapeutic strategies for hepatic fibrosis.

## 1. Introduction

Hepatic fibrosis is a histological consequence of the wound-healing process resulting from chronic liver injuries induced by various causes. Advanced fibrosis progresses to liver cirrhosis leading to various life-threatening complications and hepatocellular carcinoma [1]. During long-standing liver injuries, the activation of hepatic stellate cells (HSCs) following hepatocyte damage and the recruitment of inflammatory mediators lead to the accumulation of extracellular matrix (ECM) [2]. At this time, reactive oxygen species (ROS) are primarily generated in the mitochondria and endoplasmic reticulum of hepatocytes, leading to further hepatocyte damage that results in HSC activation and enhanced ECM production [3]. These vicious pathogenic events of involving hepatocyte damage, inflammation, ROS production, and excessive ECM accumulation can accelerate hepatic fibrosis.

Mitochondria in hepatocytes serve as the primary source of energy; however, their dysfunction is commonly associated with increased ROS production. Moreover, along with being the source of ROS, mitochondria and mitochondrial DNA (mtDNA) can suffer damage by ROS. Thus, mitochondrial ROS homeostasis is critical for preventing oxidative injury in hepatocytes [4,5]. Once mtDNA is damaged by ROS produced in mitochondria, a cascade of events culminating in apoptosis or cell death proceeds. Studies have suggested that mitochondrial dysfunction in injured hepatocytes can initiate the apoptotic pathway, leading to increased collagen production via HSC stimulation [6,7,8]. Growing evidence supports a link between mitochondrial dysfunction and liver fibrogenesis, and mitochondrial quality control-based therapy has emerged as a new therapeutic strategy. However, it remains unknown whether mitochondrial dysfunction, specifically in hepatocytes, plays a role in the fibrogenesis, or whether mediators from hepatocyte mitochondrial damage promote liver fibrosis.

High-temperature requirement protein A2 (HtrA2, also known as Omi) is a nuclear encoded serine protease that localizes in the intermembrane space of mitochondria under normal conditions, and it is released into the cytosol upon apoptosis in response to various cellular stresses [9]. The pro-apoptotic function of HtrA2/Omi protease is at least partially mediated via the binding and proteolytic removal of inhibitor of apoptosis proteins. Recent studies illustrated that HtrA2/Omi inactivation does not cause early lethality in non-neuronal tissue, unlike its effects in neuronal tissue, but it leads to increased accumulation of mtDNA deletions and premature aging in mammals [10,11]. It has also been demonstrated that HtrA2/Omi deficiency causes mtDNA damage through ROS generation and DNA mutation, which can lead to the accumulation of unfolded proteins in the mitochondria, oxidative stress, and defective mitochondrial respiration, suggesting that HtrA2/Omi is important for mitochondrial homeostasis. Furthermore, our previous studies indicated that HtrA2/Omi deficiency or point mutations in its protease domain cause mtDNA conformational changes through ROS production in cultured cells [12]. The Ser276Cys (S276C) missense mutation in HtrA2/Omi was found to be the cause of symptoms such as muscle wasting, neurodegeneration, involution of the spleen and thymus, and death by 40 days of age in mnd2 (motor neuron degeneration 2) mutant mice. In these mice, the protease activity of HtrA2/Omi is greatly reduced. 

Given that both ROS and mitochondrial dysfunction contribute to liver fibrogenesis and that hepatocyte mtDNA can exacerbate HSC activation, we hypothesized that HtrA2/Omi plays a pivotal role in liver fibrosis by modulating mitochondrial homeostasis. 

In the present study, we demonstrated that the progression of fibrosis in both animal models and patients is associated with decreased expression of HtrA2/Omi, which modulates mitochondrial function and ROS generation. The modulation of HtrA2/Omi through mitochondrial homeostasis might be a promising anti-fibrotic therapeutic approach. These findings suggest the therapeutic value of HtrA2/Omi in the treatment of liver fibrosis.

## 2. Materials and Methods

### 2.1. Clinical Samples and Animal Studies 

Five liver fibrosis tissues were obtained from patients with diagnosed chronic liver diseases who underwent liver transplantation (Seoul St. Mary’s Hospital, Seoul, South Korea) prior to 2010 and stored in liquid nitrogen. None of them had history of any treatment. In addition, three liver tissues (as controls) were also obtained from patients without viral hepatitis during surgical procedures, and they were described in a previous report [13]. All patients provided written informed consent for the storage of liver tissue samples according to the ethical guidelines of Seoul St. Mary’s Hospital in the Catholic University of Korea. Their personal information was restricted to analytical purposes. Such information is not available to the public.

All animal care and experimental protocols were conducted in accordance with the guidelines for the Care and Use of Laboratory Animals provided by the Research Supporting Center for Medical Science of the Catholic University of Korea (2016-0005-03). BALB/C and heterozygous *mnd2* (mnd2+/−) mice of the B6(Cg)-Htra2^mnd2^/J strain were purchased from Orient Bio (Seongnam, Republic of Korea) and Jackson Laboratory (stock no. 004608). Mnd2/mnd2, mnd2/+, and WT mice were obtained by crossing mnd2 heterozygous (mnd2/+) mice. The genotypes of the mice were identified via PCR-AluI-RFLP genotype analysis as previously described [14]. Mice were used when 6–8 weeks old, excluding mnd2/mnd2 mice, which were used at 3 weeks of age, and housed in a standard laboratory animal facility.

To establish an animal model of liver fibrosis, male BALB/C or mnd2/+ mice (from five to seven mice per group) were treated via intraperitoneal injections of CCl_4_ (Sigma, St. Louis, MO) as previously described [15]. Briefly, mice received CCl_4_ dissolved in mineral oil (1/4 ratio) or mineral oil alone at a dose of 0.5 mL/kg body weight twice a week for 8 weeks to induce liver fibrosis. The control group received mineral oil alone at the same time. For the preventive study, liver-targeted hydrodynamic gene delivery to the mice was performed as previously described [16,17]. In brief, saline containing 30 μg of pFLAG-HtrA2/Omi plasmid, an expression vector containing the murine HtrA2/Omi open reading frame [18], or its control plasmid was hydrodynamically injected into the liver via a catheter with temporal blood flow occlusions. The injection volume and flow rate were fixed at 5% body weight and 1 mL/s, respectively. CCl_4_ and the pFLAG-HtrA2/Omi plasmid were administered from five to seven mice in each group every 3 days for 8 weeks. The mice were sacrificed, and their livers were harvested.

### 2.2. Histological Analysis and Immunohistochemistry

Liver tissues were fixed in 3.7% buffered formalin, and then embedded in paraffin wax. The samples were cut into 3-μm sections and stained with hematoxylin & eosin (H&E) and Sirius Red (Direct Red 80, Aldrich, Milwaukee, WI) to detect collagen deposition. For immunohistochemistry, serial sections were deparaffinized and hydrated through a graded alcohol series. Antigen retrieval was performed by heating the sample in 0.01 M citrate buffer (pH 6.0) using a microwave vacuum histoprocessor (RHS-1, Milestone, Bergamo, Italy) at a controlled final temperature of 121 °C for 15 min. To block endogenous peroxide activity, the sections were quenched in 3% hydrogen peroxide in methanol and then blocked with 1% bovine serum albumin in PBS. Sections were incubated with primary antibodies against α-SMA and HtrA2/Omi diluted 1:500 in Antibody Diluent (Golden Bridge, Mukilteo, WA) at 4 °C. After washing, the peroxidase EnVision System (HRP rabbit/Mouse Envision System TM, Dakocytomation, Denmark) was applied at room temperature for 5–10 min. Peroxidase activity was detected with 3,3′-diaminobenzidine tetrachloride (DakoCytomation) and hematoxylin counterstain (DakoCytomation). The percent staining was calculated by the software of the Optimas 6.5 system.

### 2.3. TUNEL Assay

The TUNEL assay was performed using an in-situ cell death detection kit (Roche Diagnostics GmbH, Mannheim, Germany) following the manufacturer’s protocol. After staining, the sections were mounted with mounting medium with 4, 6-diamidino-2-phenylindole (DAPI; Sigma). Apoptotic cells were quantified by counting TUNEL-positive nuclei. For each sample, the number of TUNEL-positive cells was observed under a fluorescent or confocal microscope (Zeiss, Jena, Germany) and counted under ×400 magnification. Six representative fields were evaluated for each mouse in all the experimental groups. 

### 2.4. Isolation of Mouse Primary Hepatocytes and Cell Culture 

Mice were intraperitoneally anesthetized with Rompun (10 mg/kg) and Zoletil (40 mg/kg). These mice were then exsanguinated. Livers were perfused in situ through portal vein with calcium- and magnesium-free Hanks’ balanced salt solution (HBSS, Welgene, Daegu, Republic of Korea) until the firm texture was lost. After perfusion, soft liver tissue was removed and placed in a 1:1 mixture of Dulbecco’s modified Eagle’s medium and Ham’s F-12 medium (DMEM/F12, Invitrogen, Carlsbad, CA). Subsequently, the liver suspension was poured through sterile 70-μm nylon mesh (BD Sciences, San Jose, CA) and then the homogenate was centrifuged at 50 × g for 2 min. The pellet containing parenchymal cells was washed twice with DMEM/F12 containing 10% fetal bovine serum (FBS, Invitrogen). Isolated primary hepatocytes were plated onto collagen coated plates and cultured in DMEM/F12 supplemented with 10% FBS. The non-tumorigenic mouse hepatocyte cell line FL83B cells was cultured in Ham’s F-12K medium containing 10% FBS (Invitrogen), 100 μg/mL penicillin, and 0.25 μg/mL streptomycin. The LX-2 human hepatic stellate cell line (Merck Millipore, Billerica, MA; SCC064) with key features of hepatic stellate cytokine signaling and fibrogenesis was used as described previously [19]. LX-2 cells were cultured in DMEM supplemented with 10% FBS (Invitrogen), 100 μg/mL penicillin, and 0.25 μg/mL streptomycin. The cells were maintained in a humidified incubator at 37 °C with 5% CO_2_.

### 2.5. Western Blot Analysis

Protein was extracted from cell lysates using RIPA lysis buffer (10 mM Tris-HCl, pH 7.5; 10 mM EDTA; 1% NP-40; 0.1% SDS; 150 nM NaCl; 0.5% sodium deoxychloride; protease inhibitors) for western blotting. Protein extracts were heated at 100 °C for 5 min before loading followed by separation on 10% or 12% SDS-polyacrylamide gels, transfer onto nitrocellulose membranes (Schleicher & Schuell, Dassel, Germany), and blocking for 1 h at room temperature in 5% skim milk. The membranes were incubated with primary antibodies overnight at 4 °C, followed by incubation (2 h at room temperature) with HRP-conjugated secondary antibodies (Amersham Biosciences, Cardiff, UK). Target proteins were detected using an enhanced chemiluminescence system (Amersham Pharmacia Biotech, Uppsala, Sweden) according to the manufacturer’s instructions. The density of each band was analyzed using the Multi Gauge V3.0 program (Fujifilm, Tokyo, Japan).

### 2.6. Transmission Electron Microscopy (TEM)

Cells were collected and fixed with 4% paraformaldehyde and 2.5% glutaraldehyde in 0.1 M phosphate buffer, pH 7.2, at 4 °C overnight. After rinsing with 0.1 M phosphate buffer three times for 30 min each, the cells were treated with 1% osmium tetroxide in 0.1 M phosphate buffer for 1 h, dehydrated through a graded series of ethanol and acetone, embedded in Epon 812, and polymerized at 60 °C for 3 days. Ultrathin sections (60–70 nm) were prepared using an ultramicrotome (Leica Ultracut UCT; Leica Microsystems GmbH, Wetzlar, Germany). The sections on Formvar-coated slot grids were examined under a transmission electron microscope (JEM 1010; JEOL Ltd., Tokyo, Japan) operated at 60 kV. Images were recorded using a CCD digital camera (Orius SC1000; Gatan, Pleasanton, CA). All experiments were repeated three to five times to ensure reproducibility.

### 2.7. Immunofluorescence Staining

Cells were fixed with 4% paraformaldehyde for 20 min and permeabilized with 0.5% Triton X (Sigma). After washing three times each with PBS, the cells were blocked with 1% bovine serum albumin in PBS. Subsequently, the cells were incubated overnight at 4 °C with the primary antibodies. The cells were washed three times with PBS and incubated with Alexa Flour 488-labeled anti-rabbit IgG (Life Technologies, Carlsbad, CA). Nuclei were visualized by staining for 5 min with 1 μg/mL DAPI. After washing, the preparations were mounted using Kaiser’s Glycerol gelatin (Merck, Darmstadt, Germany). The fluorescence intensity of the preparations was detected using a confocal microscopy (Zeiss). Fluorescence intensity in cells per sections was measured with microscope image-analysis software (ZEN blue software) by a single investigator who was blind to sample identity. In each of four replicate experiments, 20 images were recorded, and the fluorescent cells in each image were counted (every image contained approximately 20 cells). Finally, the cell concentration was calculated from the average number of cells per image.

### 2.8. Serum Aminotransferase Activity and Hydroxyproline Determination

Hepatotoxicity was assessed by quantifying the activities of serum alanine aminotransferase (ALT) using an ALT assay kit according to the manufacturer’s protocol (Vettest 8008 Chemistry Analyzer; IDEXX Lab., UK). The accumulation of collagen in the liver tissue was determined by estimating the hydroxyproline content, an amino acid characteristic of collagen. Hydroxyproline levels in mouse livers were measured using a hydroxyproline assay kit (BioVision, Milpitas, CA) according to the manufacturer’s instructions. The results are reported as milligrams of hydroxyproline per gram of wet liver tissue.

### 2.9. Lentiviral Vector Transduction

To establish a stable HtrA2/Omi-depleted cell line, FL83B cells were infected with a mouse HtrA2/Omi specific shRNA-encoded lentivirus (Sigma; SHCLNV-NM_019752). An shRNA negative control lentiviral particle (LV-Control) was used as a negative control. To generate a stable cell line, FL83B cells were plated at a density of 1 × 10^5^ cells per 60-mm culture dish and infected overnight with five multiplicities of infection (MOI) lentiviral particles in the presence of 8 μg/mL hexadimethrine bromide (Sigma). After infection, the transduced cells were selected using 10 mg/mL puromycin (Sigma) for 2 weeks and incubated at 37 °C in a humidified incubator with 5% CO_2_. Suppression of HtrA2/Omi expression in selected cells was confirmed by western blot analysis.

### 2.10. Quantitative Real Time-PCR-Based Gene Expression

Total RNA was extracted with TRIzol reagent (Invitrogen) and treated with DNase I (Invitrogen). For first-strand cDNA synthesis, 1.5 μg of total RNA were reverse-transcribed at 42 °C for 1 h using a random hexamer primer (Applied Biosystems) and Superscript II reverse transcriptase (Invitrogen). mRNA levels were measured using SYBR Premix Ex Taq (Takara, Japan). The relative mRNA levels were quantified using the comparative ΔCT method, normalized to β-actin. Primer sequences are listed in Appendix A.

### 2.11. Mitochondrial Fractionation and mtDNA Extraction

Following cell lysis, mitochondria were prepared using a mitochondria isolation kit (Pierce Biotechnology, Inc., Rockford, IL), according to the manufacturer’s protocol. Isolation and DNase treatment of mitochondrial pellets were performed as described previously [20]. The DNase and RNase-treated mitochondrial pellet was resuspended in lysis buffer via gentle pipetting and the suspension was incubated at 37 °C for 1 h. A measure of 2 mg of proteinase K (Roche Diagnostics) was added and the lysate was incubated for 1 h at 37 °C. mtDNA was purified according to the genomic DNA extraction protocol using a DNeasy Blood & Tissue Kit (Qiagen, Santa Clarita, CA).

### 2.12. Genomic DNA Extraction and Quantitative PCR (qPCR)

Preparation of total genomic DNA from cell or liver tissue was performed using a DNeasy Blood & Tissue Kit (Qiagen, Santa Clarita, CA). Kits were used according to the manufacturer’s instructions with the inclusion of RNAse A treatment to generate RNA-free genomic DNA, and genomic DNA was eluted using sterile deionized water. Quantitative PCR (qPCR) was conducted on genomic DNA using SYBR Premix Ex Taq in triplicate for each sample.

mtDNA damage was determined as a ratio of the copy number of short mtDNA-79 bp fragments (indicative of damaged mtDNA) to the copy number of long mtDNA-230 bp fragments (indicative of undamaged mtDNA) of the mitochondrial 16S-RNA gene as previously reported (Appendix A) [21]. In addition, the mtDNA copy number was compared to determine the relative mtDNA:nDNA ratio. Primers were designed within the mitochondria NADH dehydrogenase 1 (*mt-ND1*), and cytochrome oxidase 1 (*mt-COX1*) region of the mitochondrial genome (Appendix A). The nuclear NADH dehydrogenase flavoprotein 1 (*Ndufv1*) gene was used to standardize the mtDNA copy number to the diploid chromosomal DNA content [22]. Relative gene expression was normalized to that of the single-copy nuclear *Ndufv1* gene (ΔCT) in each sample. 

### 2.13. Mitochondrial Membrane Potential and ROS Production

Cells were incubated with 2 uM CM-H_2_DCFDA (Molecular Probes/Invitrogen) resuspended in warm HBSS or HBSS alone for unstained controls for exactly 15 min. The cells were analyzed on a FACSCalibur flow cytometer (BD). The cellular subset was identified according to size and granularity. We used a mitochondria-specific dye (MitoTracker Green FM) that binds the mitochondrial membrane independently of the membrane potential, and thus, the staining intensity is considered an index of mitochondrial mass. For MitoSOX Red-based flow cytometric detection of mitochondrial superoxide, cells were then incubated with MitoSOX Red superoxide indicator (Invitrogen) for 30 min, washed, and then analyzed on a FACSCalibur. The mean channel fluorescence was converted to absolute fluorescence using an inverse log transformation and normalized to that of untreated cells or WT hepatocytes.

### 2.14. Measurement of Mitochondrial Respiration

The OCR and extracellular acidification rate of cells were measured using a Seahorse XF24 extracellular flux analyzer (Seahorse Bioscience, Billerica, MA). In brief, hepatocytes were plated on Seahorse XF 24well plates at a density of 5 × 10^4^ per well to achieve 80–90% confluency at the time of assay. Following the overnight attachment of cells, the medium was replaced with Seahorse XF medium, and the manufacturer’s protocol for the Mitostress kit was followed (Seahorse Bioscience). In this analysis, sequential injections of 1 μM oligomycin, 1 μM FCCP, and 0.5 μM rotenone/antimycin A were added to the cells to define the basal OCR, ATP-linked OCR, proton leak, maximal respiratory capacity, reserve respiratory capacity, and non-mitochondrial oxygen consumption. Results for mitochondrial respiration were normalized to the total protein content.

### 2.15. Serine Protease Activity Assy

The protease activity of HtrA2/Omi in liver sections from WT and mnd2 heterozygous (mnd2/+) mice was assayed with the substrates β-casein. Liver lysates were immunoprecipitated (IP) with HtrA2/Omi-specific polyclonal antibody. The IP complexes were incubated for the indicated times at 37 °C with β-casein as a substrate. The reaction samples were resolved by 15% SDS-PAGE, and the processing pattern of β-casein was visualized by staining with Coomassie Brilliant Blue dye (CBB). The level of the HtrA2 was analyzed by western bolt with HtrA2/Omi Ab.

### 2.16. Statistical Analysis

All results are expressed as the mean ± SEM. Comparisons between two groups were performed using two-tailed Student’s *t*-test. * *p* < 0.05, ** *P* < 0.005, *** *p* < 0.001.

Comparisons between multiple groups were performed via two-way analysis of variance (ANOVA). When ANOVA identified significant differences, individual means were compared using the post hoc Bonferroni test. Statistical analyses were performed using GraphPad Prism software 6.0.

## 3. Results

### 3.1. Mitochondrial Dysfunction Is Present in the CCl_4_-Induced Mouse Model of Liver Fibrosis

As has been reported for liver fibrosis, most forms of chronic liver diseases are associated with the accumulation of damaged mitochondria, which are responsible for abnormal ROS formation and respiratory complex alterations [23]. To investigate whether alterations in mitochondrial structure or functions in hepatocyte were associated with the progression of liver fibrosis, we established a mouse model of CCl_4_-induced liver fibrosis. As shown in Figure 1A, liver sections from the CCl_4_ group displayed a distorted architecture with extensive collagen deposition upon staining with H&E and Sirius Red. Further examination via TEM revealed obvious swelling in mitochondria, and the cristae disappeared in mouse livers during the progression of hepatic fibrosis, suggesting that the mitochondrial structure was damaged along with these fibrotic changes (Figure 1A). Based on these findings, we hypothesized that the structural alterations of mitochondria were due to mitochondrial damage and that mtDNA damage, such as damage-associated molecular patterns (DAMPs), accumulated in necrotic hepatocytes. Therefore, we performed a quantitative evaluation of damaged mtDNA and mtDNA content in CCl_4_-induced fibrotic livers. As in previous reports [21], we isolated genomic DNA, and then performed qPCR assay for two different mtDNA, namely 79 bp fragment (damaged), and 230 bp fragment (undamaged). The structurally damaged mitochondria in CCl_4_-induced fibrotic model showed an increase in the damaged mtDNA at the ratios of the 79 bp fragment and 230 bp fragment (Figure 1B). In addition, we performed qPCR assay for two different mtDNA markers, namely mt-ND1 and mt-COX1. The levels of these mtDNA markers were normalized against NADH dehydrogenase flavoprotein 1 (Ndufv1) levels to examine the relative mtDNA to nuclear DNA ratios as described previously [24]. As shown in Figure 1C, the copy numbers of *mt-ND1* and *mt-COX1* per *Ndufv1* were significantly increased by 2.2- and 2.3-fold, respectively in CCl_4_-induced fibrotic model. Based on these results, the increased damaged mtDNA and mtDNA content in CCl_4_-induced fibrotic livers can result in malfunctioning proteins and altered mtDNA replication and/or transcription efficiency. Next, we examined the expression levels of mitochondrial respiratory and complex activity-encoded genes in mouse livers during the progression of hepatic fibrosis using qRT-PCR and western blot analysis. As shown in Figure 1D, the expression levels of nuclear-encoded subunit of complex IV *ATP5A* (Complex V) and *COX5B* (Complex IV) mRNA were decreased in CCl_4_-induced fibrotic livers. Furthermore, immunoblots illustrated that the levels of subunits of ATP5A and MTCO1 (Complex IV) were significantly decreased in CCl_4_-induced fibrotic livers, whereas those of the SDHB (Complex II) subunit were unchanged (Figure 1E). Therefore, these findings indicated that the pathogenesis of liver fibrosis is associated with mitochondrial damage or mitochondrial dysfunction.

### 3.2. HtrA2/Omi Expression Is Decreased in CCl_4_-Induced Fibrotic Mice and Patients with Liver Fibrosis

We attempted to identify and characterize the role of the mitochondrial serine protease HtrA2/Omi in improving mitochondrial damage during the progression of hepatic fibrosis. We first used fibrotic mouse models to validate the association of HtrA2/Omi expression with liver fibrosis. We found that HtrA2/Omi expression was downregulated in CCl_4_-treated mouse livers via immunohistochemistry and western blotting. Staining assays revealed that HtrA2/Omi predominantly localized in the cytoplasm of hepatocytes and its levels were lower in fibrotic livers than in normal livers (Figure 2A). Likewise, immunoblotting demonstrated that HtrA2/Omi expression was markedly decreased in mouse livers during the progression of hepatic fibrosis (Figure 2B). We detected α-smooth muscle actin (α-SMA) expression as a marker of HSC activation in fibrotic liver tissues. These findings were consistent with results obtained via western blotting using liver tissue samples from patients with fibrosis and healthy controls (Figure 2C). Immunohistochemistry of HtrA2/Omi in human fibrotic liver revealed that liver tissue from patients with late-stage fibrosis (grade 4) had lower HtrA2/Omi expression than that from patients with early-stage fibrosis (grades 1–2) (Figure 2D). These results suggest that HtrA2/Omi expression is downregulated in liver fibrosis with mtDNA alterations or mitochondrial dysfunction.

### 3.3. HtrA2/Omi-Deficient Hepatocytes Cause Mitochondrial Accumulation and Structural Anomalies

To investigate the direct relationship between HtrA2/Omi downregulation and mitochondrial function in hepatocyte during liver fibrogenesis, we transfected a plasmid encoding shRNA targeting HtrA2/Omi into FL83B mouse hepatocytes, which were named lenti-shHtrA2 cells. As shown in Appendix A, HtrA2/Omi protein levels were significantly lower in lenti-shHtrA2 cells than in negative control lenti-shNC cells. We previously reported that HtrA2/Omi deficiency causes mtDNA damage through mutation and ROS generation, which can lead to mitochondrial dysfunction and consequent cell death [12]. Therefore, we examined the effect of HtrA2/Omi depletion on intracellular total ROS and mitochondria ROS (mtROS) levels in lenti-shHtrA2 hepatocytes using CM-H_2_DCFDA and MitoSox. As shown in Figure 3A,B, lenti-shHtrA2 cells produced greater amounts of mtROS despite a decrease of total ROS than lenti-shNC cells. Recent reports suggested that mitochondria play a key role in regulating cell size by affecting the balance of cell growth and proliferation through metabolic activity [25]. Therefore, we measured the distributions of cell volume by analyzing forward (FSC) and side (SSC) light scatter as a cell-size and granularity index and then measured the mitochondrial mass via staining with a mitochondria-specific dye (MitoTracker FM) and intracellular voltage-dependent anion channel (VDAC) expression in HtrA2/Omi-depleted FL83B cells. In addition to the absence of changes in the cell-size and granularity index (Appendix A), the MitoTracker signal and VDAC expression revealed that the average mitochondrial mass was not changed in HtrA2/Omi-depleted hepatocytes (Appendix A). Next, the degree of mtDNA damage induced by mtROS was measured in HtrA2/Omi-depleted hepatocyte via qPCR assay. As shown in Figure 3C, the ratios of the 79 bp fragment and 230 bp fragment were significantly increased by 1.7-fold in HtrA2/Omi-depleted hepatocyte. Furthermore, an interesting observation was that *mt-ND1* mtDNA content per *Ndufv1* in HtrA2/Omi-depleted FL83B cells was significantly increased by approximately 6-fold compared with control levels, whereas *ND1* mRNA levels were significantly decreased (Figure 3D,E). These results are consistent with previous findings that MEF cells lacking HtrA2/Omi displayed increased mtDNA levels relative to paired control cell lines [12]. Mitochondrial oxidative stress-induced mtDNA damage was associated with a decrease in mitochondrial respiration [26]. Next, we examined the effect of HtrA2/Omi depletion on mitochondrial respiration by measuring the oxygen consumption rate (OCR) using an extracellular flux analyzer. Three basal OCRs were recorded, followed by the sequential injection of oligomycin, FCCP, and antimycin A. As shown in Figure 3F, HtrA2/Omi-depleted FL83B cells displayed impaired mitochondrial respiration either under basal or maximal oxygen consumption induced by FCCP treatment. Taken together, these results indicate that HtrA2/Omi-deficient hepatocytes exhibit mitochondrial dysfunction and a concomitant elevation of mtROS levels.

### 3.4. Loss of HtrA2/Omi Protease Activity in Hepatocytes Results in the Accumulation of Dysfunctional Mitochondria and Oxidative Stress

We attempted to confirm the association of mitochondrial dysfunction with mtROS levels following the loss of HtrA2/Omi mitochondrial protease activity as well as HtrA2/Omi depletion in hepatocytes. Motor neuron degeneration 2 (mnd2)-mutant mice carry a single missense mutation (Ser276Cys) in the HtrA2/Omi gene that inactivates the protease activity of HtrA2/Omi [27]. Consistent with previous studies [28], mnd2-mutant mice exhibited striatal neuron loss; severe muscle wasting; weight loss; general decreases in the sizes of organs such as the liver, thymus, heart, and spleen; and death before 40 days of age (Appendix A). We next examined whether HtrA2/Omi mutation influences the quantity and function of mitochondria in hepatocytes as well as liver fibrogenesis. We observed mitochondrial morphology using TEM in liver tissue from wild-type (WT) or mnd2-mutant mice at postnatal day 32. Compared with the findings in WT mice, the liver tissue sections from mnd2-mutant mice appeared to have greater numbers of mitochondria and a slightly larger volume (Appendix A). In addition, the mitochondria in the liver tissue sections from mnd2-mutant mice were swollen compared with those in WT mice. These findings indicate that HtrA2/Omi mutation induces mitochondrial accumulation, mitochondrial swelling, and disruption of the cristae, and the results are similar for primary hepatocytes isolated from mnd2-mutant mouse livers (Appendix A).

Based on the abnormal mitochondrial morphology and biogenesis in the livers of mnd2-mutant mice, HtrA2/Omi mutation in hepatocytes is expected to lead to mitochondrial dysfunction. We hypothesized that pathologic changes in mitochondria caused by HtrA2/Omi mutation result from abnormalities of respiratory complex subunits. As mentioned previously, mnd2-mutant mice displayed decreases in liver size, and thus, we compared the cell-size and granularity index of primary hepatocytes isolated from WT and mnd2-mutant mouse livers at postnatal day 32 via FSC and SSC analysis (Figure 4A). FACS analysis illustrated that the cell-size and granularity index was lower for hepatocytes from mnd2 mutant mice. A previous study found that mitochondrial activity changes with cell size, resulting in allometric scaling of metabolism at the cellular level [25,29]. To determine whether HtrA2/Omi mutation in hepatocytes affects mitochondrial mass and mtROS, we assessed VDAC expression, MitoTracker Green, and MitoSOX staining in primary hepatocytes isolated from mnd2-mutant mouse livers via western blot, flow cytometry, and confocal microscopy. As shown in Figure 4B, mtROS levels to be higher in mnd2-mutant hepatocytes than in WT hepatocytes. However, no differences in mitochondrial mass by VDAC expression and MitoTracker staining were observed between hepatocytes isolated from mnd2-mutant and WT mouse livers. Furthermore, confocal microscopy revealed that mnd2-mutant hepatocytes had smaller mitochondrial areas, but higher mitochondrial fluorescence intensity for MitoSOX Red than WT hepatocytes (Figure 4C). We also measured intracellular ROS levels using the intensity of CM-H_2_DCFDA fluorescence in H_2_O_2_-treated hepatocytes. As shown in Figure 4D, intracellular ROS levels were increased in mnd2-mutant mouse hepatocytes treated with 1.5 mM H_2_O_2_. ROS levels were higher in mnd2-mutant mouse hepatocytes, and mnd2-mutant mouse hepatocytes are more sensitive to H_2_O_2_-induced oxidative stress than in WT hepatocytes. These findings suggest that the number of mitochondria per cell was not changed by mnd2 mutation in primary hepatocytes, whereas mtROS and intracellular ROS levels were elevated.

To clarify whether increased mtROS levels are involved in mitochondrial dysfunction in primary hepatocytes isolated from mnd2-mutant mouse livers, we assessed damaged mtDNA and mtDNA content using real-time qPCR. As shown in Appendix A, *mt-ND1* and *mt-COX1* were detectable at higher levels in mnd2 hepatocyte, whereas their mRNA levels were significantly lower. In addition, the ratios of the 79 bp fragment and 230 bp fragment were significantly increased by 5.5-fold in mnd2 hepatocyte. These results are consistent with those in HtrA2/Omi-deficient hepatocytes and CCl_4_-induced fibrotic livers. In addition, mitochondrial respiration (*ATP5A, COX5B*) and biogenesis (*ERRα*) genes were downregulated in mnd2-mutant hepatocytes compared with their levels in WT hepatocytes (Appendix A). These results reveal statistically significant associations between mtDNA damage and the concomitant elevation of mtROS levels in mnd2-mutant hepatocytes.

### 3.5. Loss of HtrA2/Omi Mitochondrial Protease Activity in mnd2-Mutant Mice Promotes Liver Fibrosis

The abnormal mitochondrial shape and mitochondrial dysfunction observed in HtrA2/Omi-deficient liver tissue might be closely linked to liver fibrogenesis. Next, we induced chronic liver injury in mnd2 heterozygous (mnd2/+) mice via repetitive CCl_4_ injections to observe the development of extensive bridging fibrosis and substantial collagen deposits. As shown in Appendix A, we demonstrated that the heterozygous (mnd2/+) mice for this deletion showed a 50–70% reduction in mitochondrial protease activity. The mnd2/+ mice were used because mnd2 homozygous (mnd2/mnd2) mice die before 40 days of age. After 8 weeks of CCl_4_ treatment, mnd2/+ mice displayed significant hepatic fibrosis, as demonstrated by quantification of Sirius-red positive area (Appendix A), compared with the findings in WT mice. Moreover, the concentration of hydroxyproline in mnd2/+ mouse livers was also increased after CCl_4_ injection compared with the levels in WT mice (Appendix A). These results suggest that loss of HtrA2/Omi mitochondrial protease activity in mnd2-mutant mice promotes liver fibrosis by increasing mtDNA damage and mtROS levels.

### 3.6. HtrA2/Omi Deficient Hepatocyte Derived-mtDNA Induces Liver Fibrogenesis

Based on the aforementioned results, we demonstrated a direct link between increased mtDNA damage in HtrA2/Omi-deficient or HtrA2/Omi-mutated hepatocytes and liver fibrogenesis. HtrA2/Omi was predominantly expression in hepatocyte, but there is little information available regarding the effects of potential paracrine stimulation by hepatocyte-derived mtDNA or mtROS on HSC activation either in vivo or in culture. Therefore, we hypothesized that the accumulation of damaged mtDNA in hepatocyte may serve as DAMPs to link the HSC activation. We isolated damaged mtDNA from HtrA2/Omi-deficient or HtrA2/Omi-mutated hepatocytes and examined whether mtDNA can induce HSC activation as DAMP molecules. Forty-eight hours after adding the same concentration (500ng) of mtDNA extracted from HtrA2/Omi-deficient hepatocytes, upregulation of the mRNA transcripts for *collagen 1* and *α-SMA* was observed in inactivated LX-2 cells with serum-free medium (Figure 5A). Furthermore, LX-2 cells underwent morphological changes in response to damaged mtDNA treatment, as shown in stained images (Figure 5B). Consistent with the upregulation of *collagen 1* and *α-SMA* transcripts in LX-2 cell treated with mtDNA from HtrA2/Omi-mutated hepatocytes, *collagen 1* and *α-SMA* expression was upregulated in LX-2 cell treated with mtDNA from primary hepatocytes isolated from mnd2-mutant mice (Figure 5C,D). These results, together with the previously mentioned data, strongly suggested that the accumulation of damaged mtDNA due to a loss of HtrA2/Omi in hepatocyte is associated with liver fibrosis through crosstalk with the activation of HSC.

### 3.7. Restoration of HtrA2/Omi Expression Rescues CCl_4_-Induced Liver Fibrosis and Reverses Mitochondrial Dysfunction in Hepatocyte

Because HtrA2/Omi expression is downregulated during hepatic fibrogenesis, it was expected that HtrA2/Omi plays a protective role in our CCl_4_-induced liver fibrosis model. To further confirm the protective role of HtrA2/Omi, we examined the effects of the hydrodynamic gene delivery of pFLAG-HtrA2/Omi in the CCl_4_-induced liver fibrosis model. In a standard CCl_4_-induced mouse model of liver fibrosis, serum ALT activity was significantly changed in mice treated with CCl_4_ twice weekly for 8 weeks. The hydrodynamic injection of HtrA2/Omi attenuated the elevation of ALT activity in CCl_4_-treated mice (Figure 6B). Subsequent experiments were performed to analyze liver histological alterations occurring in response to HtrA2/Omi injection in CCl_4_-treated mouse livers. Semiquantitative IHC detection of HtrA2/Omi expression cells confirmed a transient increase in the HtrA2/Omi-injected liver than in control CCl_4_- treated livers (Figure 6C). Intriguingly, pFLAG-HtrA2/Omi administration inhibited the development of hepatic fibrosis, as confirmed by H&E and Sirius red staining (Figure 6A). Quantification indicated that the Sirius red-positive area was smaller (by 5.8%) in fibrotic livers from mice injected with pFLAG-HtrA2/Omi plasmids than in livers from CCl_4_-treated mice (Figure 6D). The hydroxyproline content was significantly lower in livers treated with pFLAG-HtrA2/Omi (1.1 μg/mg, *p* < 0.01) than in control CCl_4_- treated livers (2.17 μg/mg) (Figure 6E). Hepatic HtrA2/Omi expression significantly decreased the hepatic hydroxyproline level. Immunohistochemical staining using α-SMA antibody illustrated that strong α-SMA expressions was limited to scarred areas in the CCl_4_ treatment group, but pFLAG-HtrA2/Omi administration decreased α-SMA expression in fibrotic areas (Figure 6A). As previous studies reported that HtrA2/Omi directly contributes to apoptosis [9], we evaluated apoptosis using the terminal deoxynucleotidyl transferase dUTP nick-end labeling (TUNEL) assay in HtrA2/Omi-injected tissues. In the CCl_4_-treated group, 76% of cells were TUNEL-positive, versus 67% of cells in the HtrA2/Omi-injected group (Appendix A). Consistent with the percent of TUNEL-positive apoptotic cells, we found that cleaved caspase 3 levels were reduced in the HtrA2/Omi-injected group compared with those in CCl_4_-treated group (Appendix A). These results suggest that HtrA2/Omi expression appears to reverse, or at least prevent, further progression of liver fibrosis.

Furthermore, to determine whether HtrA2/Omi expression could reverse mitochondrial dysfunction induced by CCl_4_ in hepatocyte, we compared the mitochondrial ultrastructure and mtDNA content in the fibrotic livers of mice injected with pFLAG-HtrA2/Omi plasmids. Additional analyses revealed that mitochondria more frequently had a normal structure within hepatocyte from the HtrA2/Omi-injected group (Figure 7A). As shown in Figure 7B, the ratios of the 79 bp fragment and 230 bp fragment were decreased in the HtrA2/Omi-injected group than in the CCl_4_-treated control group. The *mt-ND1* and *mt-COX1* mtDNA contents were significantly lower in the HtrA2/Omi-injected group than in the CCl_4_-treated control group (Figure 7C). Conversely, there was no difference in *ATP5A* and *COX5B* mRNA levels between the HtrA2/Omi-injected and CCl_4_-treated groups (Appendix A), but MnSOD and CuZnSOD expression in antioxidant enzymes was restored by HtrA2/Omi treatment (Figure 7D). Taken together, these data suggested that HtrA2/Omi has an important role in maintaining mitochondrial homeostasis that might decrease vulnerability to liver injury and the development of liver fibrosis.

## 4. Discussion

In this study, we demonstrated that mitochondrial dysfunction in hepatocytes is closely linked to hepatic fibrosis, and HtrA2/Omi might play a critical role in preventing of hepatic fibrogenesis through regulating mitochondrial homeostasis. Mitochondria are vital intracellular organelles that are altered in response to cellular stress and metabolic changes in hepatocytes. Oxidative stress is considered a key accelerator of liver fibrosis, and ROS produced by hepatocytes promote HSCs activation, resulting in excessive ECM deposition [30,31]. Furthermore, increased ROS production is associated with mitochondrial dysfunction in hepatocytes during liver damage [23,32], and subsequent mitochondrial dysfunction leads to oxidative stress and changes in mtDNA damage and calcium homeostasis, resulting in an energy crisis that can eventually lead to hepatocyte death. In the present study, we evaluated the alteration in the mitochondrial structure or function in a mouse model of CCl_4_-induced liver fibrosis. In addition, we demonstrate that downregulation of HtrA2/Omi expression in CCl_4_-induced liver fibrosis has a major role in modulating mitochondrial function and ROS generation in vivo and in vitro. Studies have shown that mitochondrial dysfunctions associated with HtrA2/Omi is a key causative factor inducing cell death in various chronic pathological conditions in numerous human diseases, such as neurodegeneration and cardiovascular diseases [33,34]. However, the relationship between mitochondrial dysfunction induced by HtrA2/Omi and chronic liver disease is unclear. Our analysis revealed that HtrA2/Omi-deficient and HtrA2/Omi-mutant hepatocytes have considerable reductions in mitochondrial electron transport chain activity and altered mitochondrial ultrastructural organization. However, we found that mitochondrial mass remained unchanged and the production of total ROS and mtROS increased in HtrA2/Omi deficient hepatocytes. These observations are consistent with previous studies demonstrating that HtrA2/Omi deficiency is involved in the accumulation of intracellular ROS [12,35,36], suggesting that mitochondrial damage in chronic liver injury leads to oxidative stress and changes in mtDNA damage that can eventually result in hepatocyte death. Although there was no change in the mitochondrial mass, the reason for the increased mtDNA contents in HtrA2/Omi deficient or mutation hepatocytes was that the destruction of mtDNA, which exists as a supercoiled form, caused morphological transfer to mtDNA of relaxed circular and linear forms. Accumulating evidence has demonstrated an association that the accumulation of relaxed and linearized mtDNA was used as a relatively good template for DNA amplification.

A number of recent studies reported correlations between elevated levels of intracellular or circulating mtDNA and various human diseases [37,38,39]. A study by Zhang et al. found that traumatic injury induces the release of mitochondrial DAMPs such as formyl peptides and mtDNA into the circulation, and these circulating mitochondrial DAMPs activate multiple inflammatory signal pathways through a specific receptor, toll-like receptor 9 (TLR9) [40]. TLR9 is in intracellular compartments and recognizes unmethylated cytosine phosphate guanine-containing DNA. Similarly, serum mtDNA levels are significantly higher in patients where acetaminophen-induced liver injury is significantly higher than in healthy controls, suggesting that the extent of mtDNA release into the circulation can be measured as a mechanistic biomarker of mitochondrial damage in patients with liver injury [41,42,43]. However, studies on how damaged mtDNA accumulated in hepatocytes or mtDNA released from hepatocytes affect non-parenchymal liver cells such as HSCs and cause liver fibrogenesis are still limited. Although we did not extract the damaged mtDNA released from HtrA2 /Omi deficient and HtrA2/Omi mutated hepatocytes, it showed the possibility that the accumulation of damaged mtDNA in hepatocytes can induce HSC activation in the form of DAMP molecules. These results were the first to demonstrate that HtrA2/Omi expression regulates mtDNA damage and mitochondrial homeostasis in hepatocytes during liver fibrogenesis.

Initially, the roles of HtrA2/Omi in apoptosis, mitochondrial protein folding quality control, and cell survival were investigated [9]. A recent study reported that HtrA2/Omi regulates autophagy and inflammasome signaling by preventing prolonged accumulation of the inflammasome adaptor ASC [44]. In addition, Michell et al. found that expression of the pro-apoptotic protein Bcl-2 in the liver protected against CCl_4_ induced-mitochondrial dysfunction and -oxidative stress in hepatocytes [8]. Therefore, we observed that, by protecting the mitochondria, HtrA2/Omi could restore mitochondrial structure and mitochondrial function, and subsequent play a pathophysiological role in the liver fibrogenesis in CCl_4_-induced liver fibrosis model. By hydrodynamics-based gene transfer, the overexpression of HtrA2/Omi leads to antifibrotic effects in CCl_4_-induced liver fibrosis mice model through decreasing collagen accumulation and enhancing anti-oxidative activity by modulating mitochondrial homeostasis in the liver. These results suggest that suppression of HtrA2/Omi expression promotes hepatic fibrogenesis by modulating mitochondrial ROS generation, and these novel mechanistic insights involving the regulation of mitochondrial homeostasis by HtrA2/Omi may be of importance for developing new therapeutic strategies for hepatic fibrosis.

## Figures and Tables

**Figure 1 cells-08-01119-f001:**
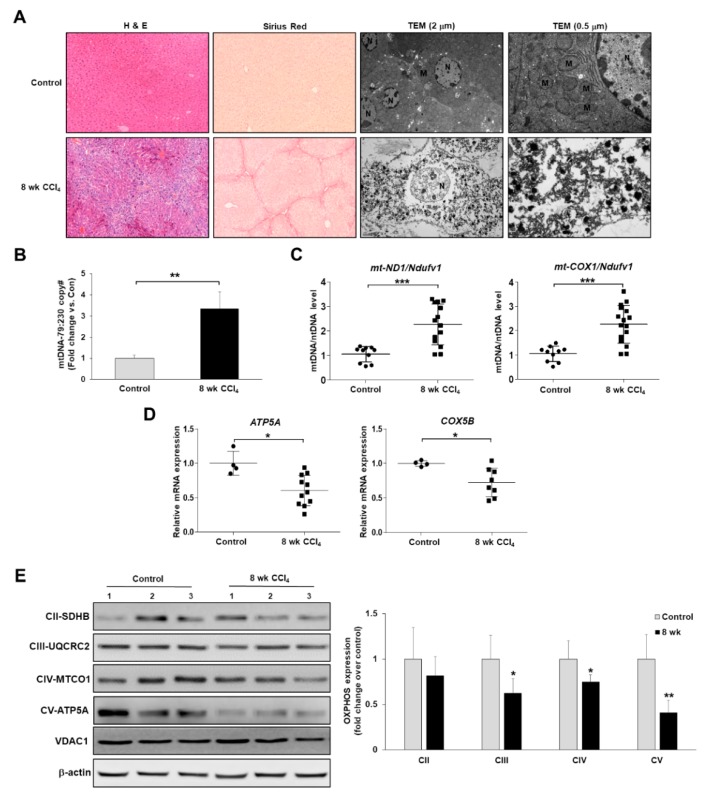
Morphological and functional abnormalities in mitochondrial are present in the mouse model of CCl_4_-induced liver fibrosis. Male BALB/c mice were injected intraperitoneally with CCl_4_ biweekly for 8 weeks to establish a hepatic fibrosis mouse model. (**A**) Representative images of H&E and Sirius red staining (original magnification, X200) of liver sections. TEM analysis of hepatocyte showing nucleus (N) and mitochondria (M) (scale bar = 2 & 0.5 μm). (**B**) Damaged mtDNA levels of the 79 bp fragment (damaged) and 230 bp fragment (undamaged) in gDNA isolated from livers of mice were assessed by qPCR. Bars represent mean copy number ratios of mtDNA-79:230, normalized to 18S levels. (**C**) Analysis of mtDNA content to obtain the mtDNA/nDNA ratio. gDNA isolated from livers of mice was analyzed by qPCR for the indicated genes. (**D**) Total RNA isolated from livers of mice was analyzed by qRT-PCR for the indicated genes. (**E**) Western blot analysis of representative subunits of OXPHOS complexes expression.

**Figure 2 cells-08-01119-f002:**
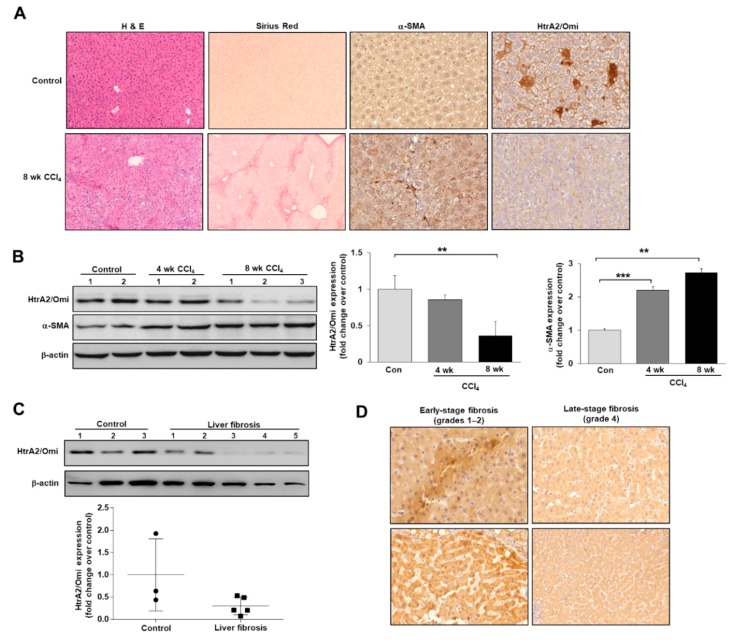
HtrA2/Omi expression is downregulated in CCl_4_-induced liver fibrosis and human fibrotic livers. (**A**) Representative images of H&E, Sirius red (original magnification, X200) and immunohistochemistry staining (X400) of liver sections. (**B**) Western blot analysis of HtrA2/Omi, α-SMA and β-actin expression. (**C**) The protein levels of HtrA2/Omi in liver tissue from patients with fibrosis and healthy controls, as evaluated by western blotting. (**D**) Immunohistochemical staining of HtrA2/Omi in human fibrotic liver (X200).

**Figure 3 cells-08-01119-f003:**
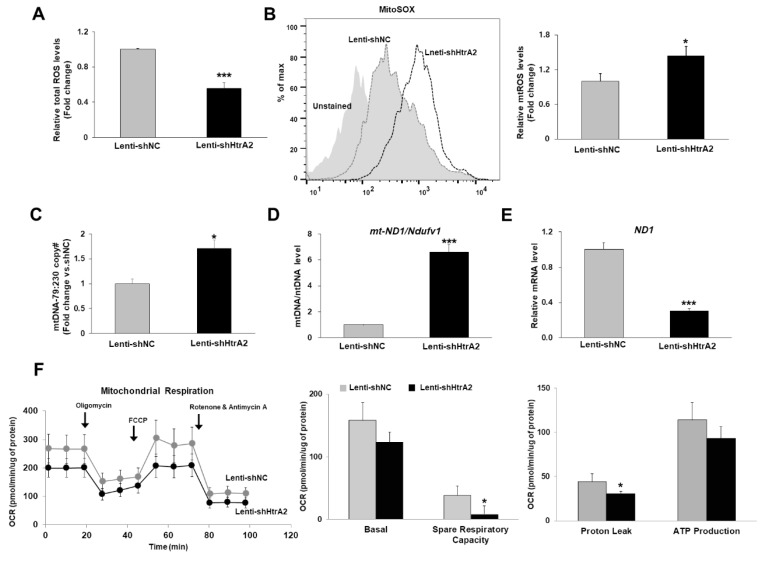
Lentivirus-mediated HtrA2/Omi depletion in hepatocyte lead to impaired mitochondrial function and metabolism. (**A**) Intracellular ROS levels were measured by FACS-analysis using CM-H_2_DCFDA staining. (**B**) Mitochondrial superoxide level was measured by FACS-analysis using MitoSOX red staining. The fluorescence mean intensity of MitoSOX red per cell was quantified. (**C**) Damaged mtDNA levels of the 79 bp fragment (damaged) and 230 bp fragment (undamaged) in gDNA isolated from cells were assessed by qPCR. Bars represent mean copy number ratios of mtDNA-79:230, normalized to 18S levels. (**D**) Real time qPCR analysis of *mt-ND1* DNA normalized to nuclear *Ndufv1* DNA. (**E**) *ND1* mRNA expression, as determined by qRT-PCR. (**F**) OCR measurements were obtained using an extracellular flux analyzer. Results for mitochondrial respiration were normalized to the total protein content.

**Figure 4 cells-08-01119-f004:**
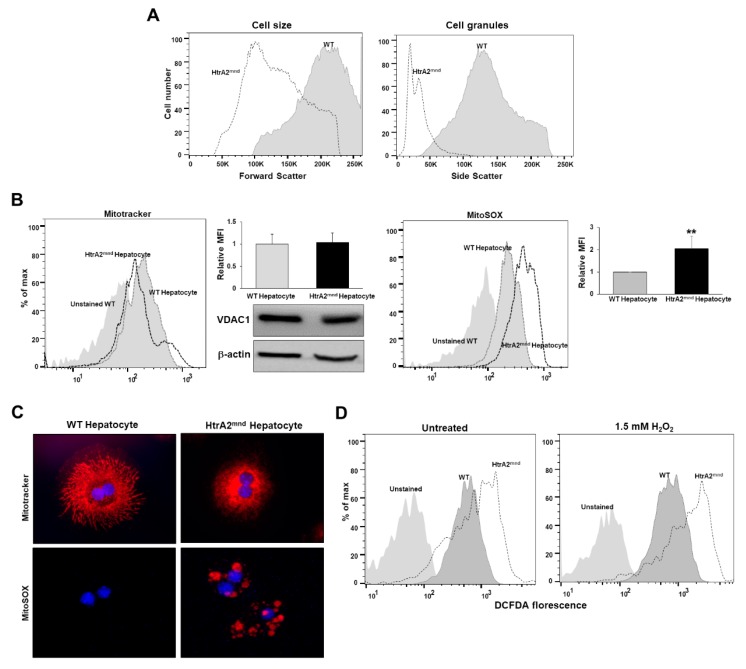
Loss of HtrA2/Omi in hepatocyte results in the accumulation of dysfunctional mitochondria and oxidative stress. (**A**) Exemplary flow cytometry plot and histogram of showing populations of primary hepatocyte in light scatters. (**B**) Mitochondrial morphology and superoxide levels in isolated hepatocyte were measured by FACS-analysis using MitoTracker green and MitoSOX red staining. The fluorescence mean intensity per cell was quantified. Western blotting for VDAC, mitochondrial mass proteins in WT and mnd2-mutant hepatocytes, with β-actin as a loading control. (**C**) Confocal microscopy of Mitotracker and MitoSOX red staining in primary hepatocyte. Nuclei were stained blue by DAPI. (**D**) Intracellular ROS levels in primary hepatocyte were measured by FACS-analysis using CM-H_2_DCFDA staining.

**Figure 5 cells-08-01119-f005:**
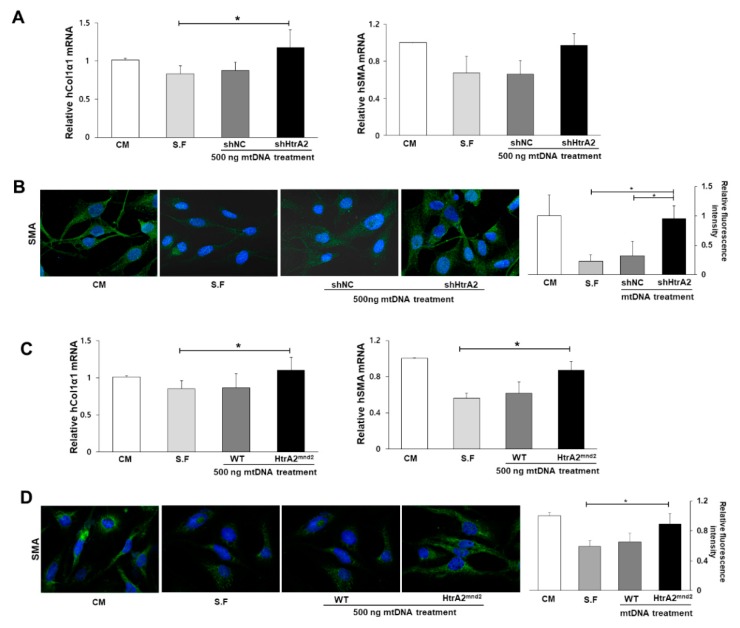
Loss of HtrA2/Omi mitochondrial protease activity promotes liver fibrosis. (**A**,**C**) Collagen 1 and α-SMA mRNA expression in LX-2 cell treated with the same concentration (500 ng) of hepatocyte derived-mtDNA, as determined by qRT-PCR. (**B**,**D**) Immunofluorescence staining for α-SMA (green) performed in LX-2 cell treated with hepatocyte derived-mtDNA. Nuclei were stained with DAPI (blue). The relative intensity measurement of immunofluorescence is shown as histogram for α-SMA. Original magnification, X400. CM: cultured LX-2 cells in completed media condition; S.F.: cultured LX-2 cells in serum free media condition.

**Figure 6 cells-08-01119-f006:**
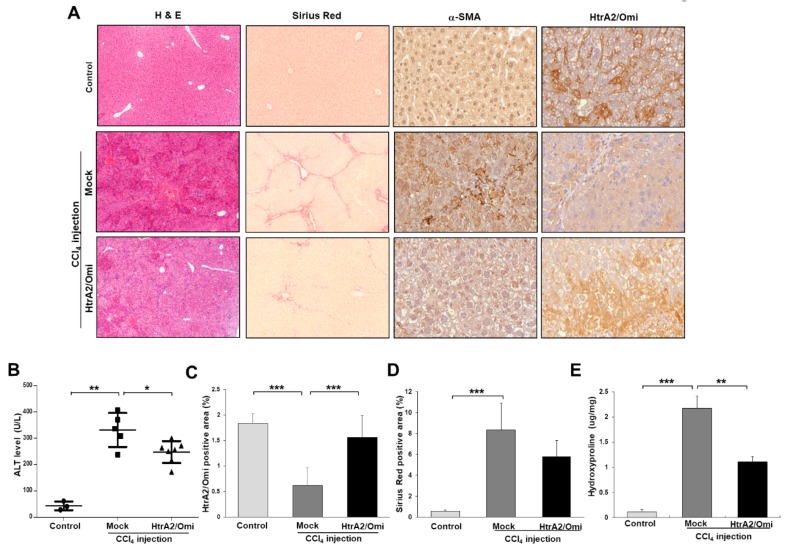
HtrA2/Omi expression in mouse model of CCl_4_-induced liver fibrosis alleviates liver fibrosis by protecting hepatocytes damage. HtrA2/Omi expression in CCl_4_-treated mice following the hydrodynamic tail vein injection of 30 µg HtrA2/Omi-encoding plasmid DNA (*n* = 7) compared with the mock control group (*n* = 5) over 8 weeks at 4-day intervals. Hepatic fibrosis was induced by injection of CCl_4_ two times per week for 8 weeks. (**A**) Representative images of H&E, Sirius red (original magnification, X200) and immunohistochemistry staining (X400) of liver sections. (**B**) Effects of HtrA2/Omi expression on serum ALT. (**C**) IHC quantification of HtrA2/Omi positivity and (**D**) semi-quantitative analysis of Sirius red staining. (**E**) The hydroxyproline content in mouse livers.

**Figure 7 cells-08-01119-f007:**
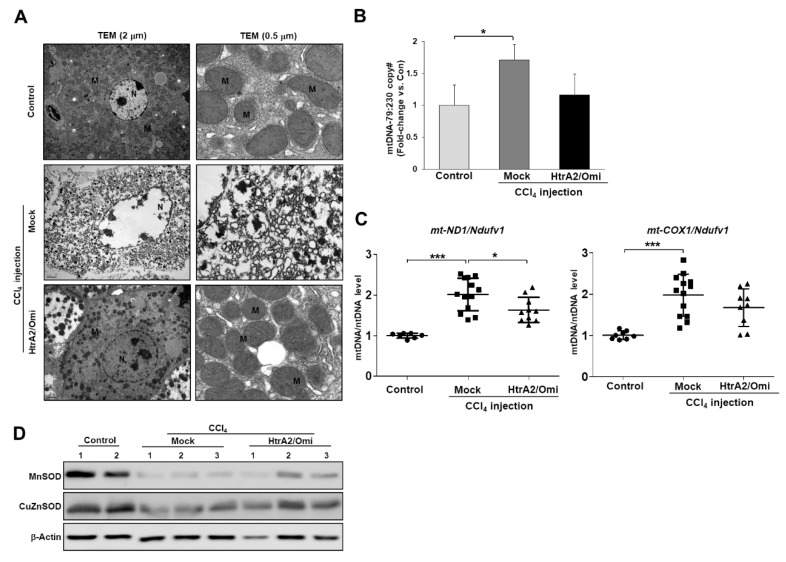
HtrA2/Omi expression in CCl_4_-induced liver fibrosis protects mitochondrial damage of hepatocyte. (**A**) TEM analysis of hepatocyte showing nucleus (N) and mitochondria (M) (scale bar = 2 & 0.5 μm). (**B**) Damaged mtDNA levels of the 79 bp fragment (damaged) and 230 bp fragment (undamaged) in gDNA isolated from liver tissue were assessed by qPCR. Bars represent mean copy number ratios of mtDNA-79:230, normalized to 18S levels. (**C**) qPCR analysis of *mt-ND1* and *mt-COX1* DNA normalized to nuclear *Ndufv1* DNA. (**D**) Western blot analysis of MnSOD and CuZnSOD expression in liver tissue from pFLAG-HtrA2/Omi treated fibrotic mice compared to mock vector treated fibrotic mice.

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
