# Peer review of "Serine Protease HtrA2/Omi Deficiency Impairs Mitochondrial Homeostasis and Promotes Hepatic Fibrogenesis via Activation of Hepatic Stellate Cells"

_cells, 2019, doi:10.3390/cells8101119_

Round 1

Reviewer 1 Report

In this manuscript by Hur et al., the authors have characterized the role of HtrA2/Omi in regulating hepatic fibrogenesis. They found that HtrA2/Omi controls mitochondrial functional homeostasis. Down-regulation of HtrA2/Omi impairs mitochondrial OXPHOS activity and promotes hepatic fibrogenesis. This manuscript is overall clear and easy to follow. However, the authors should address the following questions. 

1.    In line 311, the authors concluded that “damaged” mtDNAs accumulate in CCl4-induced mice. However, experiments in this manuscript don’t directly show that the accumulating mtDNAs are damaged. The corresponding experiments in Fig 1 B, C, D showed that, in CCl4-induced mice, mtDNA level increases while the mitochondrial mRNA/protein level decreases. These phenotypes don’t lead to the conclusion that mtDNAs are genetically mutated. For example, these phenotypes may also be caused by malfunction of mitochondrial transcription/translation. If the authors want to conclude that the accumulating mtDNAs are “damaged”, they need to sequence the mtDNA. Otherwise, the authors should restate the conclusions from these experiments. Similar imprecise conclusions can be found in multiple places in the manuscript, including but not limited to line 447, line 479, line 462, line 474 and the conclusions based on Figure 3 C, D.

2.    In Figure 5, when LX-2 cells are treated with the mtDNAs extracted from HtrA2/Omi-deficient cells, collagen and SMA mRNAs are upregulated. The authors concluded that “HtrA2/Omi deficient hepatocyte derived-mtDNA induces liver fibrosis”. However, when LX-2 cells are treated with the mtDNAs extracted from control cells (bar 3 in Figure 5A), collagen and SMA mRNAs also increased. Therefore, one possibility is that collagen and SMA mRNAs will be upregulated when LX-2 cells are treated with any mtDNAs, independent of whether the mtDNAs being damaged or not. And the difference in bar 3 and bar 4 can be explained by the fact that HtrA2/Omi-deficient cells contain more copies of mtDNA than control cells (as shown in Fig 2C). The authors should address this caveat.

3.    In Figure 5, it is not clear how mtDNAs caused the upregulation of the collagen and SMA mRNAs. Are the mtDNAs being imported? If so, how do the imported mtDNAs influence the transcription of specific genes? The authors should discuss these obvious questions.

4.    In multiple places of this manuscript (like line 539, line 306), the authors mentioned that they measured the mtDNA level through qRT-PCR. It should be noted that qRT-PCR is used to measure RNA levels rather than DNA levels. The authors actually have accurately described qRT-PCR in the method section 2.10.

5. In Figure 6, the authors restored the HtrA2/Omi expression level through hydrodynamic injection. The authors should show the protein level of HtrA2/Omi in hydrodynamic injected cells, for example, using westernblots. The column 4 of Figure 6A only showed the protein level in a small number of cells and lacks quantification.

6.    Error bar is missing in bar 3 Figure 6D

Author Response

Dear

Thank you for your consideration of our manuscript entitled “Serine protease HtrA2/Omi deficiency impairs mitochondrial homeostasis and promotes hepatic fibrogenesis via activation of hepatic stellate cellsby Wonhee Hur et al. (Cells-557021) for publication in Cells.

We have carefully considered the reviewers’ suggestions and have revised the manuscript accordingly.

Point-by-Point Responses to Reviewer I

Q-1. In line 311, the authors concluded that “damaged” mtDNAs accumulate in CCl4-induced mice. However, experiments in this manuscript don’t directly show that the accumulating mtDNAs are damaged. The corresponding experiments in Fig 1 B, C, D showed that, in CCl4-induced mice, mtDNA level increases while the mitochondrial mRNA/protein level decreases. These phenotypes don’t lead to the conclusion that mtDNAs are genetically mutated. For example, these phenotypes may also be caused by malfunction of mitochondrial transcription/translation. If the authors want to conclude that the accumulating mtDNAs are “damaged”, they need to sequence the mtDNA. Otherwise, the authors should restate the conclusions from these experiments. Similar imprecise conclusions can be found in multiple places in the manuscript, including but not limited to line 447, line 479, line 462, line 474 and the conclusions based on Figure 3 C, D.

- As suggested by the reviewer, we have included the new results of damaged mtDNA contents. Previous studies (Alcohol. 2018. pii: S0741-8329(18)30181 /PLoS One. 2013;8(5):e64413 / BJU International, 2008;102:628) reported the mtDNA damage was determined as a ratio of the copy number of short mtDNA-79 bp fragments (indicative of damaged mtDNA) to the copy number of long mtDNA-230 bp fragments (indicative of undamaged mtDNA). Based on these reports, we have included some results on the ratio of damaged mtDNA (see Figure 1B, 3C, 7B, Supplementary Fig. S2F; page 6, line 268-271 & page 7, line 326-331 & page 10, line 402-406 & page 12, line 483-484 & page 15, line 573-575 in revision).

Q-2.  In Figure 5, when LX-2 cells are treated with the mtDNAs extracted from HtrA2/Omi-deficient cells, collagen and SMA mRNAs are upregulated. The authors concluded that “HtrA2/Omi deficient hepatocyte derived-mtDNA induces liver fibrosis”. However, when LX-2 cells are treated with the mtDNAs extracted from control cells (bar 3 in Figure 5A), collagen and SMA mRNAs also increased. Therefore, one possibility is that collagen and SMA mRNAs will be upregulated when LX-2 cells are treated with any mtDNAs, independent of whether the mtDNAs being damaged or not. And the difference in bar 3 and bar 4 can be explained by the fact that HtrA2/Omi-deficient cells contain more copies of mtDNA than control cells (as shown in Fig 2C). The authors should address this caveat.

- Thank you for your comments. As described in the previous study (Gut 2005;54:142), the human stellate cell line LX-2 cell was generated by spontaneous immortalization in low serum conditions. The phenotype of LX-2 is most similar to that of “activated” HSC in vivo. Despite this activated phenotype, LX-2 cell line can be quiesced under low serum condition. For these reasons, we performed the experiments that the activated LX-2 under was reversed under serum free condition (S.F; presentive of inactivated HSC), and thus examined the effects of HSC activation following treatment with mtDNA isolated from hepatocytes. In this process, there could be experimental error under the influence of various experimental conditions such as serum-free and cell number. Thus, we added the new results through at least 3 times repeated experiments (see Figure 5A-5D; page 13, line 512-519 in revision).

Q-3. In Figure 5, it is not clear how mtDNAs caused the upregulation of the collagen and SMA mRNAs. Are the mtDNAs being imported? If so, how do the imported mtDNAs influence the transcription of specific genes? The authors should discuss these obvious questions.

- As mentioned in introduction, liver fibrosis occurs as a wound-healing scar response following chronic liver inflammation including alcoholic liver disease, non-alcoholic steatohepatitis (NASH) and hepatitis B & C. During long-standing liver injuries, HSCs activation following hepatocyte damage and the recruitment of inflammatory mediators lead to the accumulation of extracellular matrix (ECM), such as fibrillar collagen. Recent studies reported that intracellular or extracellular mtDNA levels increased in chronic liver disease in mice and humans (J Clin Invest. 2012;122(4):1574 / Nature. 2010;464(7285):104). The elevated mtDNA may act as damage-associated molecular patterns (DAMPs), which can initiate an inflammatory response through a specific receptor, toll-like receptor 9 (TLR9) (Hepatology. 2007;46:1509 / Free Radic Biol Med. 2011;51(2):424). TLR9 is in intracellular compartment and recognizes unmethylated cytosine phosphate guanine-containing DNA. By binding to TLR9, mtDNA can also activate the MAP kinases. Studies have shown that the accumulated hepatocyte-derived mtDNA from NASH livers had a greater ability to activate TLR9, resulting in induction of fibrogenic responses. Subsequently activated TLR9 elevated  mRNA levels of TGF-β and collagen type I in HSCs through the MyD88/TAK signaling pathway, and therefore promotes hepatic fibrogenesis (J Clin Invest. 2016;126(3):859 / Hepatology. 2007;46:1509 / Int J Cancer. 2018; 142(1): 81). Based on these previous studies, we showed that the damaged mtDNA isolated from HtrA2/Omi deficient and HtrA2/Omi-mutated hepatocytes could induce HSC activation in the form of DAMP molecules. Although our study was not able to demonstrate how hepatocyte-derived mtDNA is involved in hepatocyte-HSC cross talk via the TLR9 signaling pathway, our results showed that damaged mtDNA dependent on HtrA2/Omi expression in hepatocyte is involved to fibrogenesis by HSC activation. Thus, we consider that further research regarding this issue is needed. As suggested by the reviewer, we have included some sentence in the discussion section (see page 17, line 624-635 in revision).

Q-4. In multiple places of this manuscript (like line 539, line 306), the authors mentioned that they measured the mtDNA level through qRT-PCR. It should be noted that qRT-PCR is used to measure RNA levels rather than DNA levels. The authors actually have accurately described qRT-PCR in the method section 2.10.

- As recommended, we have corrected them in the revised manuscript (see page 5, line 246 & page 5, line 262- page 6, line 267 & page 7, lines 331 & page 10, lines 402 in revision).

Q-5. In Figure 6, the authors restored the HtrA2/Omi expression level through hydrodynamic injection. The authors should show the protein level of HtrA2/Omi in hydrodynamic injected cells, for example, using western blots. The column 4 of Figure 6A only showed the protein level in a small number of cells and lacks quantification.

- As suggested by the reviewer, we have included the quantitative result of HtrA2/Omi expression levels in Figure 6C (see page 14, line 542-543 in revision). In general, after the hydrodynamic injection of target DNA, the expression level of DNA plasmid peaks at 2-3 days and gradually decrease thereafter. For this reason, we were not able to show the expression of HtrA2/Omi expression levels at 1 wk after last DNA injection.

Q-6. Error bar is missing in bar 3 Figure 6D.

- As recommended, we have corrected it in the revised Figure 6 (see page 15 in revision)

We hope that our revised manuscript has satisfactorily addressed the concerns of the Reviewers. We express our utmost appreciation and gratitude for your valuable remarks.

Yours sincerely, 

Reviewer 2 Report

Review of the manuscript: Serine protease HtrA2/Omi deficiency impairs mitochondrial homeostasis and promotes hepatic fibrogenesis via activation of hepatic stellate cells Wonhee Hur, Byung Yoon Kang, Sung Min Kim, Gil Won Lee, Jung-Hee Kim, Min-Kyung Nam, Hyangshuk Rhim and Seung Kew Yoon.

In this manuscript, Hur and coworkers describe a link between mitochondrial dysfunction in hepatocytes, hepatic fibrosis and the activity of the Serine protease HtrA2/Omi. They evaluated the alteration in the mitochondrial structure and  function in a mouse model of CCl4-induced liver fibrosis and the also correlate those findings with the espression on the HtrA2/Omi protein in vivo and in vitro. However, the main critical point of this work is the assumption that an increase in mtDNA copy number linked to altered expression of some mitochondrial genes is enough to demonstrate a damage in mtDNA itself. The assumption that damaged mtDNA is present in fibrosis mouse models and in cell lines, and that is linked to HtrA2/Omi function is therefore not  sufficiently solid.  

I have the following comments - Methods:

·         Number of mice and human liver samples used is not reported in some experiments

·         The Mnd2 mouse model is not adequately described in the introduction.

·         For the preventive study, W. Hur and coworkers chose liver-targeted hydrodynamic gene delivery. It would be appropriated to justify this choice and explain why they did not perform gene delivery with AAV vectors or similar.

·         The H&E and Sirius Staining do not look at 200x and 400x magnification, but most probably 20x and 40x; can the authors check this?

·         Section 2.11 (Mitochondrial fractionation and genomic DNA extraction) is not clearly stated. The authors describe a method to isolate mitochondria and subsequently treat them with DNAse I to remove genomic DNA and obtain pure mtDNA (as indicated in ref 20, Higuchi, Y.; Linn, S. Purification of all forms of HeLa cell mitochondrial DNA and assessment of damage to it caused by hydrogen peroxide treatment of mitochondria or cells. J Biol Chem 1995, 270, 7950-7956), but it seems that this is a method to obtain genomic DNA (as indicatred in the title), and to perform subsequent mtDNA quantification via real-time PCR (section 2.12). This section should be revised and written more accurately. Moreover, in section 2.12, the authors chose NDUFV1  gene for mtDNA quantification, but the same choice was for normalisation of gene expression assays… ? In my opinion these 2 sections will benefit of more accurate rewriting.

Results – 3.1.

·         The authors describe a 2.2-2.3-fold increase in mtDNA copy number in CCl4-treated livers via quantification of mtND1, mtCOXI versus NDUFV1 gene by Real-Time PCR. However, they conclude that the accumulated mtDNA is damaged. A general increase in mtDNA copy number is simply an index of mitochondrial mass (together with other markers, i.e. citrate synthase activity) that could indicate mitochondrial proliferation in response to an insult (in this case, CCl4 treatment). However, this does not indicate that the mtDNA is damaged and should be proven by appropriate experiments (sequencing or gel shift migration).

·         The authors examined the expression levels of ATP5A and COX5B and wrote “As shown in Fig.1C, the expression level of mtDNA-encoded ATP5A and COX5B mRNA were decreased in CCl4-induced fibrotic livers”. Unfortunately, COX5B and ATP5a are nuclear-encoded subunit of complex IV and complex V, respectively. Morover, the data appear not to be statistically significant  - can the authors add the statistic data?

·         I would add a blot against VDAC1/Porin as mitochondrial loading control to check the mitochondrial mass in figure 1D.

·         The conclusion (row 319-320) that the fibrosis is associated with mitochondrial DNA alteration is inappropriated.

Results – 3.3.

·         The authors show in Suppl. Figure S1C that the size of the lenti-ShHtrA2 cells is unchanged compared to controls. However, I would add a quantification of the signal rather than the graph, as the lenti-ShHtrA2 cells seem to have both lower “cell size” and “cell granules” indexes.

·         Again, the authors inappropriately measure “mtDNA damage” by measuring mtDNA copy number (figure 3C and rows 369-376). Although the increase in mtDNA copy number per cells can be a compensatory mechanism of impaired mitochondrial function, this does not prove the presence of damaged mtDNA.  ND1 mRNa levels again do not prove, in my opinion, the presence of mtDNA damage. This should be demonstrated with different experiments (as the authors did in ref12).

·         OCR measured by Seeahorse XFe24 extracellular flux analyser seems not to be decreaed in lenti-ShHtrA2 cells compared to control; the authors should include statistic tests to verify whether the differences observed in figure 3E are significant. Moreover, the Oxygen Consumption Rate is measured as pmol/min/ug of protein; the title of the y bar on the graphs are incorrected.

Results – 3.4.

·         In the characterization of mnd2 mutant mice, the authors performed TEM to evaluate mitochondrial morphology. Although  there is an apparent increase in the mitochondrial mass, the conclusion that HtrA2/Omi mutation leads to mitochondrial accumulation would benefit of more appropriate experiments (western blot for VDAC1/Porin, measurement of CS activity). In fact, the  mitotracker staining in cultured hepatocytes from mnd2-mutant livers do not show significant differences compared to wt hepatocytes (Fig. 4B). An appropriate quantification of mitochondrial mass by adding more mitochondrial markers is highly recommended.

·         Please add the results of the t-test to figure S2-F to confirm a significant decrease in the transcript levels.

Results – 3.5.

·         It’s not clear to me why the authors chose to hexamine liver fibrosis in heterozygous mnd2+/- mice rather than homozygous mnd2-/- at the same age in which they performed the characterization (section 3.4)

·         The authors conclude that loss of HtrA2/Omi mitochondrial protease activity in mnd2-mutant mice promotes liver fibrosis by increasing mtDNA damage and mtROS levels, but in my view, this is inappropriate. The mnd2+/- mice should not have a loss of protease activity but possibly a 50% reduction of enzymatic activity, has this been demonstrated +/- tissues? Is there any papers that describe that? Again, the mtDNA damage has not been proven.

Results – 3.6.

·         In this section, Wonhee Hur and coworkers describe the effect of damaged mtDNA on gene expression of human hepatic cell line Lx2; however, the experimental design is confused and the methods are not decribed so avoiding the interpretation of the results. Although some recent papers describe how mtDNA can be effectively transfer to cells/tissue via extracellular vescicles, the authors here should clearly described and validated the effectiveness of their treatment in the delivery of mtDNA within cells. Moreover, the “morphological changes” of LX-2 cells described in Fig. 5A are unclear.

In conclusion, although the authors describe some alteration in mitochondrial function in a mouse model of CCl4-induced fibrosis that could be possibly link to the function of HtrA2/Omi, the assumption that mtDNA damage and mitochondrial electron transport chain activity are related to this protein is not sufficiently proven. The final link between mtDNA, DAMPs and Htra2/Omi expression is confused and not adequately presented.

Author Response

Dear

Thank you for your consideration of our manuscript entitled “Serine protease HtrA2/Omi deficiency impairs mitochondrial homeostasis and promotes hepatic fibrogenesis via activation of hepatic stellate cellsby Wonhee Hur et al. (Cells-557021) for publication in Cells.

We have carefully considered the reviewers’ suggestions and have revised the manuscript accordingly.

Point-by-Point Responses to Reviewer II

Q-1. Number of mice and human liver samples used is not reported in some experiments

- As suggested by the reviewer, we have included the number of mice and human liver biopsy samples (see page 3, line 134-135, 144 in revision).

Q-2. The Mnd2 mouse model is not adequately described in the introduction.

- As suggested by the reviewer, we have added some sentences in the introduction and result section (see page 2, line 102-105 & page 11, line 431-432 & page 13, line 492-497 in revision).

Q-3. For the preventive study, W. Hur and coworkers chose liver-targeted hydrodynamic gene delivery. It would be appropriated to justify this choice and explain why they did not perform gene delivery with AAV vectors or similar.

- There are various viral and non-viral gene delivery system for gene over-expression or knockdown in cells or tissues. Among them, viral vector-mediated gene delivery exhibits the most effective therapeutic results, including adenovirus, adeno-associated virus, lentivirus, and so on. However, many studies have reported that its application in vivo as a standard treatment is still controversial due to the lethal immune response and carcinogenesis caused by the virus vector (Cancer Gene Ther. 2002; 9:979 / N. Engl. J. Med. 2003; 348:255). The course of liver fibrosis or advanced cirrhosis, regardless of its aetiology, is complicated by fibrosis/cirrhosis-associated immune dysfunction and impairs the homeostatic role of the liver in the systemic immune response (Front Pharmacol. 2017;8:591 / Cell. Mol. Immunol. 2016; 13:267). In contrast, non-viral gene delivery system is being extensively studied in terms of safety or biohazard for use of clinical application. For this reason, we have performed plasmid DNA-based hydrodynamic gene delivery methods due to its advantage in safety and versatility in vivo, even though gene transfer efficiency is lower. Thus, as previously described, we have hydrodynamically injected HtrA2/Omi plasmids under optimal conditions which showed high efficiency of gene transfer and minimal DNA degradation.

Q-4. The H&E and Sirius Staining do not look at 200x and 400x magnification, but most probably 20x and 40x; can the authors check this?

- We checked as suggested by the reviewer. Our magnification is correct. This probably looks like this when converted to a PDF file.

Q-5. Section 2.11 (Mitochondrial fractionation and genomic DNA extraction) is not clearly stated. The authors describe a method to isolate mitochondria and subsequently treat them with DNAse I to remove genomic DNA and obtain pure mtDNA (as indicated in ref 20, Higuchi, Y.; Linn, S. Purification of all forms of HeLa cell mitochondrial DNA and assessment of damage to it caused by hydrogen peroxide treatment of mitochondria or cells. J Biol Chem 1995, 270, 7950-7956), but it seems that this is a method to obtain genomic DNA (as indicatred in the title), and to perform subsequent mtDNA quantification via real-time PCR (section 2.12). This section should be revised and written more accurately. Moreover, in section 2.12, the authors chose NDUFV1 gene for mtDNA quantification, but the same choice was for normalisation of gene expression assays? In my opinion these 2 sections will benefit of more accurate rewriting.

- We have edited the manuscript according to the reviewer’s comment (see page 5, line 246-252 & 262-267 in revision).

Q-6. The authors describe a 2.2-2.3-fold increase in mtDNA copy number in CCl4-treated livers via quantification of mtND1, mtCOXI versus NDUFV1 gene by Real-Time PCR. However, they conclude that the accumulated mtDNA is damaged. A general increase in mtDNA copy number is simply an index of mitochondrial mass (together with other markers, i.e. citrate synthase activity) that could indicate mitochondrial proliferation in response to an insult (in this case, CCl4 treatment). However, this does not indicate that the mtDNA is damaged and should be proven by appropriate experiments (sequencing or gel shift migration).

- As suggested by the reviewer, we have included the new results of damaged mtDNA contents. Previous studies (Alcohol. 2018. pii: S0741-8329(18)30181 /PLoS One. 2013;8(5):e64413 / BJU International, 2008;102:628) reported the mtDNA damage was determined as a ratio of the copy number of short mtDNA-79 bp fragments (indicative of damaged mtDNA) to the copy number of long mtDNA-230 bp fragments (indicative of undamaged mtDNA). Based on these reports, we have included some results on the ratio of damaged mtDNA (see Figure 1B, 3C, 7B, Supplementary Fig. S2F; page 6, line 268-271 & page 7, line 326-331 & page 10, line 402-406 & page 12, line 483-484 & page 15, line 573-575 in revision).

Q-7. The authors examined the expression levels of ATP5A and COX5B and wrote “As shown in Fig.1C, the expression level of mtDNA-encoded ATP5A and COX5B mRNA were decreased in CCl4-induced fibrotic livers”. Unfortunately, COX5B and ATP5a are nuclear-encoded subunit of complex IV and complex V, respectively. Morover, the data appear not to be statistically significant  - can the authors add the statistic data?

- As suggested by the reviewer, we used wrong words in the manuscripts and have corrected them in the revised manuscript (see Figure 1D; page 7, line 340-341 in revision).

Q-8. I would add a blot against VDAC1/Porin as mitochondrial loading control to check the mitochondrial mass in figure 1D.

- As suggested by the reviewer, we have included a new Western blot result (see Figure 1E; page 8, line 347 in revision).

Q-9. The conclusion n (row 319-320) that the fibrosis is associated with mitochondrial DNA alteration is inappropriated.

- We have edited the sentence according to the reviewer’s comment (see page 7, line 344-346 in revision).

Q-10. The authors show in Suppl. Figure S1C that the size of the lenti-ShHtrA2 cells is unchanged compared to controls. However, I would add a quantification of the signal rather than the graph, as the lenti-ShHtrA2 cells seem to have both lower “cell size” and cell granules” indexes.

-  As suggested by the reviewer, we have added the quantitative result of cell size” and “cell granules” indexes in Supplementary Fig. S1C. The cell-size and granularity index seems to have no correlation between lenti-ShHtrA2 and shNC hepatocyte.

Q-11. Again, the authors inappropriately measure “mtDNA damage” by measuring mtDNA copy number (figure 3C and rows 369-376). Although the increase in mtDNA copy number per cells can be a compensatory mechanism of impaired mitochondrial function, this does not prove the presence of damaged mtDNA.  ND1 mRNa levels again do not prove, in my opinion, the presence of mtDNA damage. This should be demonstrated with different experiments (as the authors did in ref12).

- As suggested by the reviewer, we have included some results on the ratio of damaged mtDNA (see Figure 1B, 3C, 7B, Supplementary Fig. S2F; page 6, line 268-271 & page 7, line 326-331 & page 10, line 402-406 & page 12, line 483-484 & page 15, line 573-575 in revision).

Q-12. OCR measured by Seahorse XFe24 extracellular flux analyser seems not to be decreaed in lenti-ShHtrA2 cells compared to control; the authors should include statistic tests to verify whether the differences observed in figure 3E are significant. Moreover, the Oxygen Consumption Rate is measured as pmol/min/ug of protein; the title of the y bar on the graphs are incorrected.

- As suggested by the reviewer, we have included the statistic tests and corrected the title of y bar. (see Figure 3F; page 10, line 416 in revision).

Q-13. In the characterization of mnd2 mutant mice, the authors performed TEM to evaluate mitochondrial morphology. Although there is an apparent increase in the mitochondrial mass, the conclusion that HtrA2/Omi mutation leads to mitochondrial accumulation would benefit of more appropriate experiments (western blot for VDAC1/Porin, measurement of CS activity). In fact, the  mitotracker staining in cultured hepatocytes from mnd2-mutant livers do not show significant differences compared to wt hepatocytes (Fig. 4B). An appropriate quantification of mitochondrial mass by adding more mitochondrial markers is highly recommended.

- As suggested by the reviewer, we have included a new Western blot result (see Figure 4B; page 11, line 457-459 in revision). In Figure 4B, we demonstrated that no differences in mitochondrial mass by VDAC expression and MitoTracker staining were observed between hepatocytes isolated from mnd2-mutant and WT mouse livers. These results are consistent with those reported in Exp Cell Res (2010 Apr 15; 316(7): 1213).

Q-13. Please add the results of the t-test to figure S2-F to confirm a significant decrease in the transcript levels.

-  As suggested by the reviewer, we have added the results of the t-test in Supplementary Fig. S2G.

Q-14. It’s not clear to me why the authors chose to hexamine liver fibrosis in heterozygous mnd2+/- mice rather than homozygous mnd2-/- at the same age in which they performed the characterization (section 3.4)

- The loss of HtrA2/Omi protease activity causes the neuromuscular disorder of the mnd2 (motor neuron degeneration 2) mutant mice (Nature. 2003;425(6959):721). These mice develop multiple defects including neurodegeneration with parkinsonian features and exhibit striatal neuron loss, severe muscle wasting, weight loss and death before 40 days of age. Therefore, we used mnd2 heterozygous (mnd2/+) mice in this study because mnd2 homozygous (mnd2/mnd2) mice die before 40 days of age. As suggested by the reviewer, we have added some sentences on mnd2 mice in the introduction and results section. (see page 2, line 102-105 & page 11, line 431-432 & page 13, line 492-497 in revision).

Q-15. The authors conclude that loss of HtrA2/Omi mitochondrial protease activity in mnd2-mutant mice promotes liver fibrosis by increasing mtDNA damage and mtROS levels, but in my view, this is inappropriate. The mnd2+/- mice should not have a loss of protease activity but possibly a 50% reduction of enzymatic activity, has this been demonstrated +/- tissues? Is there any papers that describe that? Again, the mtDNA damage has not been proven.

- As mentioned in the response above, the mouse mnd2 mutation as the missense mutation Ser276Cys in the protease domain of HtrA2/Omi exhibits the reduction of HtrA2/Omi Protease activity. Our finding is consistent with previous study by Jones JM., et al. (Nature. 2003;425(6959):721). In this previous study, to determine the effect on protease activity, HtrA2/Omi was immunoprecipitated from liver and assayed with 35S-labelled b-casein as substrate. They showed that no b-casein cleaving activity was detectable in the immunoprecipitates from mnd2 tissues. Their results showed that the protease activity of mnd2/+ was reduced by more than 50%, which is in consistent with our new results (see Supplementary Fig. S3A; page 13, line 492-497 in revision). Furthermore, as suggested by the reviewer, we have included the new results of damaged mtDNA in HtrA2/Omi-mutant hepatocytes (see Supplementary Fig. S3F; page 12, line 483-484 in revision).

Q-16. In this section, Wonhee Hur and coworkers describe the effect of damaged mtDNA on gene expression of human hepatic cell line Lx2; however, the experimental design is confused and the methods are not decribed so avoiding the interpretation of the results. Although some recent papers describe how mtDNA can be effectively transfer to cells/tissue via extracellular vescicles, the authors here should clearly described and validated the effectiveness of their treatment in the delivery of mtDNA within cells. Moreover, the “morphological changes” of LX-2 cells described in Fig. 5A are unclear.

- As mentioned in introduction, liver fibrosis occurs as a wound-healing scar response following chronic liver inflammation including alcoholic liver disease, non-alcoholic steatohepatitis (NASH) and hepatitis B & C. During long-standing liver injuries, HSCs activation following hepatocyte damage and the recruitment of inflammatory mediators lead to the accumulation of extracellular matrix (ECM), such as fibrillar collagen. Recent studies reported that intracellular or extracellular mtDNA levels increased in chronic liver disease in mice and humans (J Clin Invest. 2012;122(4):1574 / Nature. 2010;464(7285):104). The elevated mtDNA may act as damage-associated molecular patterns (DAMPs), which can initiate an inflammatory response through a specific receptor, toll-like receptor 9 (TLR9) (Hepatology. 2007;46:1509 / Free Radic Biol Med. 2011;51(2):424). TLR9 is in intracellular compartment and recognizes unmethylated cytosine phosphate guanine-containing DNA. By binding to TLR9, mtDNA can also activate the MAP kinases. Studies have shown that the accumulated hepatocyte-derived mtDNA from NASH livers had a greater ability to activate TLR9, resulting in induction of fibrogenic responses. Subsequently activated TLR9 elevated mRNA levels of TGF-β and collagen type I in HSCs through the MyD88/TAK signaling pathway, and therefore promotes hepatic fibrogenesis (J Clin Invest. 2016;126(3):859 / Hepatology. 2007;46:1509 / Int J Cancer. 2018; 142(1): 81). Based on these previous studies, we showed that the damaged mtDNA isolated from HtrA2/Omi deficient and HtrA2/Omi-mutated hepatocytes could induce HSC activation in the form of DAMP molecules. Although our study was not able to demonstrate how hepatocyte-derived mtDNA is involved in hepatocyte-HSC cross talk via the TLR9 signaling pathway, our results showed that damaged mtDNA dependent on HtrA2/Omi expression in hepatocyte is involved to fibrogenesis by HSC activation. Thus, we consider that further research regarding this issue is needed. As suggested by the reviewer, we have included some sentence in the discussion section (see page 17, line 624-635 in revision). Furthermore, as suggested by the reviewer, we have included the relative intensity measurement of immunofluorescence for α-SMA in Fig.5B & 5D (see Figure 5B & 5D; page 13, line 512-519 in revision).

We hope that our revised manuscript has satisfactorily addressed the concerns of the Reviewers. We express our utmost appreciation and gratitude for your valuable remarks.

Yours sincerely, 

Reviewer 3 Report

Dear authors,

I read your manuscript reporting that the serine protease HtrA2 deficiency impairs mitochondrial homeostasis and promotes hepatic fibrogenesis with great interest.

While the results using four different approaches (CCl4 induced liver fibrosis, knock-down, catalytic-dead mutation, transfection of "damaged mtDNA") are largely consistent indicating a role of the serine protease in maintaining mitochondrial function thereby preventing liver fibrosis, the data set still needs some supporting evidence as outlined below under "main comments".

My main comments are:

to apply the same set of assays to the three main experimental models, please include (i) collagen staining for the S276C catalytic-dead cell line and the down-regulation cell line, (ii) please measure mtROS and intracellular ROS for the CCl4 model as well as the ND1 mRNA levels for the CCl4 model

Please include the data on collagen 1 and alpha-SMA expression in LX-2 cells treated with mtDNA obtained from mnd2 mutant mice (page 12, line 477)

Figure 5, I am not sure how significant the difference between shNC (undamaged mtDNA) and shHtrA2 (damaged mtDNA) is given the hight of the error bars (the significance test should be between shNC and shHtrA2 and not between the latter and SF (serum free)). To strengthen the claim that the "damaged mtDNA" really induces expression of col1alpha and SMA, please include a Western blot for both proteins with all controls and a cell staining for collagen 1.

Please include a data set on pFLAG-HtrA2/Omi expression in the S276C mutant cell line (Fig 7)

Minor comments

please explain the rational of the CCl4 model in section 3.1

In figure legend 1 it refers only to total RNA (B&C) although mtDNA levels are measured for mtND1

Please include a CoxB5 Western if possible in Fig 1D. I think MTCO1 levels are not going down at the 8 wk time point.

Please explain to the reader why you think intracellular ROS is going down in the down-regulation cell line but not in the catalytic-dead cell line

Page 10, line 422, please explain why the mitochondrial mass remains unchanged while the volume is increasing?

in the manuscript the term "damaged mtDNA is used" it is however not defined what the authors mean with this term. Physical damage of the mtDNA is not shown, only mtDNA from damaged mitochondria was isolated.

Explain the significance of hydroxyproline and serum alanine aminotransferase to the reader

please indicate the location of the mitochondria in Fig 1A (8 wks CCL4 and Fig 7A CCl4 Mock)

Author Response

Dear

Thank you for your consideration of our manuscript entitled “Serine protease HtrA2/Omi deficiency impairs mitochondrial homeostasis and promotes hepatic fibrogenesis via activation of hepatic stellate cellsby Wonhee Hur et al. (Cells-557021) for publication in Cells.

We have carefully considered the reviewers’ suggestions and have revised the manuscript accordingly.

Point-by-Point Responses to Reviewer III

Q-1. Please include the data on collagen 1 and alpha-SMA expression in LX-2 cells treated with mtDNA obtained from mnd2 mutant mice (page 12, line 477)

- As suggested by the reviewer, we have added the new results of collagen 1 and α-SMA transcripts in LX-2 cell treated with mtDNA isolated from HtrA2/Omi-mutated hepatocytes in Fig. 5A & 5B (see Figure 5C & 5D; page 13, line 512-519 in revision).

Q-2. Figure 5, I am not sure how significant the difference between shNC (undamaged mtDNA) and shHtrA2 (damaged mtDNA) is given the hight of the error bars (the significance test should be between shNC and shHtrA2 and not between the latter and SF (serum free)). To strengthen the claim that the "damaged mtDNA" really induces expression of col1alpha and SMA, please include a Western blot for both proteins with all controls and a cell staining for collagen 1.

- Thank you for your comments. As described in the previous study (Gut 2005;54:142), the human stellate cell line LX-2 cell was generated by spontaneous immortalization in low serum conditions. The phenotype of its cells is most similar to that of “activated” in vivo. Despite this activated phenotype, LX-2 lines can be quiesced by low serum condition. For these reasons, we performed the reversion of the activated LX-2 under serum free condition (S.F; presentive of inactivated HSC) and then examined the effects of HSC activation following treatment with mtDNA isolated from hepatocytes. In this process, already activated LX-2 cell could have experimental errors under the influence of various experimental conditions such as serum-free and cell number. Thus, we were included the new results through repeated experiments. Furthermore, we have included the relative intensity measurement of immunofluorescence for α-SMA in Fig.5B & 5D (see Figure 5A-5D; page 13, line 512-519 in revision).

Q-3. Please include a data set on pFLAG-HtrA2/Omi expression in the S276C mutant cell line (Fig 7)

- Thank you for good comment. For isolation of S276C mutant primary hepatocytes, mnd2 / mnd2 mice obtained by crossing mnd2 heterozygous (mnd2 / +) mice are required. However, there was not enough time for obtaining these mice to proceed with the experiment. As suggested by the reviewer, we will further our study on the mitochondrial function and antioxidants mechanism etc. associated with HtrA2 expression in HtrA2/Omi-deficient or mutated hepatocytes.

Q-4. please explain the rational of the CCl4 model in section 3.1

- Many studies reported that experimental liver fibrosis in mice can be induced by surgical intervention (bile duct ligation), genetic manipulation of fibrosis-related genes (Mdr2 knockout mice) or application of hepatotoxins (Fibrogenesis Tissue Repair 2013;6:19 / BMC Gastroenterol. 2010;10:79 / J. Clin. Invest. 1994;94:2481 / Current Pathobiology Reports 2014;2(4):143). In particular, the single or repeated administration of carbon tetrachloride (CCl4) has become one of the most commonly used experimental models for inducing toxin-mediated liver fibrosis. We used this CCl4-induced liver fibrosis mice model as cited in ACSnano (2010; 4(6):3005) and other papers (Biomaterials 2011;32:4951 / Gut and liver 2016; 11(1) / Journal of Cellular Biochemistry 2016;9999:1).

Q-5. In figure legend 1 it refers only to total RNA (B&C) although mtDNA levels are measured for mtND1

- We have added the legend of the Figure 1 according to the reviewer’s comment (see page 20, line 772-781  in revision).

Q-6. Please include a CoxB5 Western if possible in Fig 1D. I think MTCO1 levels are not going down at the 8 wk time point.

- As suggested by the reviewer, we have included the new representative results of OXPHOS complexes expression (see Figure 1E in revision).

Q-7 & Q-8. Please explain to the reader why you think intracellular ROS is going down in the down-regulation cell line but not in the catalytic-dead cell line & Page 10, line 422, please explain why the mitochondrial mass remains unchanged while the volume is increasing?

- As shown in Supplemental Fig.2A, the size of liver in mnd2-mutant mice is small, and the cell-size and granularity index were lower in hepatocytes of mnd2 mutant mice compared to hepatocyte of WT mice. Smaller cell size has been reported to be associated with changes in mitochondrial function which was in line with our results. A number of previous papers have suggested that this change in mitochondria function may incur changes in metabolic activity at the cellular level and oxidative stress. We totally agree with your important comment and that we should explore further regarding this issue on relationship between cell size, mitochondrial mass and ROS in the future.

Q-9. in the manuscript the term "damaged mtDNA is used" it is however not defined what the authors mean with this term. Physical damage of the mtDNA is not shown, only mtDNA from damaged mitochondria was isolated.

- As suggested by the reviewer, we have included the new results of damaged mtDNA contents. Previous studies (Alcohol. 2018. pii: S0741-8329(18)30181 /PLoS One. 2013;8(5):e64413 / BJU International, 2008;102:628) reported the mtDNA damage was determined as a ratio of the copy number of short mtDNA-79 bp fragments (indicative of damaged mtDNA) to the copy number of long mtDNA-230 bp fragments (indicative of undamaged mtDNA). Based on these reports, we have included some results on the ratio of damaged mtDNA (see Figure 1B, 3C, 7B, Supplementary Fig. S2F; page 6, line 268-271 & page 7, line 326-331 & page 10, line 402-406 & page 12, line 483-484 & page 15, line 573-575 in revision).

Q-10. Explain the significance of hydroxyproline and serum alanine aminotransferase to the reader

- Hydroxyproline is one of the most amino acids present in collagen following hydroxylation of proline moiety. Determination of hydroxyproline provides useful information in the diagnosis and prognosis of liver fibrosis caused by collagen metabolic disorders. Furthermore, alanine aminotransferase (ALT) are hepatic enzymes that are released into the bloodstream from damaged hepatocyte and upraised in the blood before the clinical signs and symptoms of liver diseases occurrence. Serum ALT levels were determined to assess liver function or hepatotoxicity. As suggested by the reviewer, we have included some sentence on these information (see page 5, line 228-231  in revision).

Q-11. please indicate the location of the mitochondria in Fig 1A (8 wks CCL4 and Fig 7A CCl4 Mock)

- As suggested by the reviewer, we have indicated the location of nucleus (N) and mitochondria (M) in TEM images and added the annotation in the Figure 1A & 7A legend (see Figure 1A & 7A; page 8, line 351-352  & page 16, lines 584 in revision).

We hope that our revised manuscript has satisfactorily addressed the concerns of the Reviewers. We express our utmost appreciation and gratitude for your valuable remarks.

Yours sincerely, 

Reviewer 4 Report

Reviewing report

The manuscript submitted by Wonhee Hur and collaborators entitled “Serine protease HtrA2/Omi deficiency impairs mitochondrial homeostasis and promotes hepatic fibrogenesis via activation of hepatic stellate cells” reports the putative role of Serine protease HtrA2/Omi in the hepatic fibrogenesis process via mitochondrial dysfunction. In their studies, authors examined how Serine protease HtrA2/Omi deficiency affects mitochondrial homeostasis in hepatocyte during the development of hepatic fibrogenesis. In their study, authors tried to demonstrate that Serine protease HtrA2/Omi expression is considerably decreased in liver tissues from the CCL4-induced liver fibrotic mice model and from patients with liver cirrhosis. The stemming results of the study are really interesting and relevant, but it misses however, several things to clarify the role of Serine protease HtrA2/Omi in liver fibrogenesis via mitochondrial dysfunction and to make stronger their work. The major and minor points are the following:

Major comments:

In their experiments on clinical samples, authors don’t give some precisions on the pharmacological treatments for the patients. Did authors make different groups in function of the pharmacological treatments? How many samples were used for the experiments? Do samples come from patients with the same liver pathology or not?

In their study, authors establish an animal model of liver fibrosis by intraperitoneal injections of CCL4 during several weeks. This model was used for their different experiments versus a control. What is the control used in these experiments? A sham group with intraperitoneal injections of saline solution during the same period? Could authors clarify this point?

In their morphological and functional abnormalities in mitochondrial experiments, authors show a decrease of CIV-MTCO1 expression in the 8 weeks CCL4 mouse model of liver fibrogenesis. The blot showed in figure 1D does not show really a decrease of this expression. It seems to be very similar than the CII-SDHB complex which is not modify. Could authors explain this confused results? Could authors precise the number of animals and the method used to quantify the expression level in these experiments?

In their experiments to check the downregulation of the HtrA2/Omi expression in the CCL4 mouse model, authors show the decrease of the HtrA2/Omi expression during the 8 weeks of treatment. It appears to decrease after 4 weeks of treatment (figure 2B on the blot), could authors explain this latency time? The HtrA2/Omi expression was also evaluated in liver tissue from patient with healthy and fibrosis liver. Why the number n of sample analysis is different between this two conditions? Basically, for all of their experiments authors have to use an identical number n of samples / experiments in their groups.

In their experiments to check the αSMA expression in the CCL4 mouse model, authors show an increase of this expression during the CCL4 treatment versus control. This protein is also up regulated during heart fibrosis after ischemia. What is the consequence of the increase of the αSMA expression on the liver fibrosis in regard of the heart fibrosis which is a similar phenomenon?

In their experiments in figure 4, authors show the consequences of the loss of the HtrA2/Omi in hepatocyte on mitochondria function and oxidative stress. They show that the loss of the HtrA2/Omi promotes an increase of the pro oxidant compounds as superoxide anion. What is the consequences of the loss of the HtrA2/Omi if the cells are treated with an anti-oxidant compound as the alpha tocopherol (vitamin e)?

In their experiments in figure 7, authors show that rescue of HtrA2/Omi expression in CCL4-induced liver fibrosis mouse model protects mitochondrial damage of hepatocyte. The fibrosis is associated to the extracellular matrix production in the tissue with a scare fibrotic tissue formation. Could authors explain why there is not significant difference between mock and HtrA2/Omi in CCL4 injection group for expression level of mt-COX1/Ndufv1? Why authors did not do a collagen dosage in their different experimental conditions?

Minor comments:

Authors used the Rompun and the Zoletil to anaesthetize their mice. Why did they use these anesthetic agents instead of classical agents such as xylazine / Ketamine?

Authors give some informations about data but these data were not shown (line 477). I think they have to show these datas in supplemental figure.

In their experiments in figure 5, authors show the relative expression of hSMA and hColα1 mRNA. I am not sure to understand clearly the statistical analysis on the histograms and I am not sure of the significance between the different conditions. Could authors clarify this point?

As mentioned before, some experiments have a different number of n, it will be better if this number is equal for each condition.

After the list of references, the legend of the figure 1 is missing.

The figure 1 is not correct in the top left.

Author Response

Dear 

Thank you for your consideration of our manuscript entitled “Serine protease HtrA2/Omi deficiency impairs mitochondrial homeostasis and promotes hepatic fibrogenesis via activation of hepatic stellate cellsby Wonhee Hur et al. (Cells-557021) for publication in Cells.

We have carefully considered the reviewers’ suggestions and have revised the manuscript accordingly.

Point-by-Point Responses to Reviewer IV

Q-1. In their experiments on clinical samples, authors don’t give some precisions on the pharmacological treatments for the patients. Did authors make different groups in function of the pharmacological treatments? How many samples were used for the experiments? Do samples come from patients with the same liver pathology or not?

- As suggested by the reviewer, we have included some more detailed sentence about clinical samples (see page 2, line 117-121 in revision). We obtained five liver fibrosis tissues from patients with diagnosed chronic liver diseases who underwent liver transplantation (Seoul St. Mary’s Hospital, Seoul, South Korea) prior to 2010 and stored in liquid nitrogen. None of them had history of any treatment.

Q-2. In their study, authors establish an animal model of liver fibrosis by intraperitoneal injections of CCL4 during several weeks. This model was used for their different experiments versus a control. What is the control used in these experiments? A sham group with intraperitoneal injections of saline solution during the same period? Could authors clarify this point?

- Many studies reported that experimental liver fibrosis in mice can be induced by surgical intervention (bile duct ligation), genetic manipulation of fibrosis-related genes (Mdr2 knockout mice) or application of hepatotoxins (Fibrogenesis Tissue Repair 2013;6:19 / BMC Gastroenterol. 2010;10:79 / J. Clin. Invest. 1994;94:2481 / Current Pathobiology Reports 2014;2(4):143). In particular, the single or repeated administration of carbon tetrachloride (CCl4) has become one of the most commonly used experimental models for inducing toxin-mediated liver fibrosis. As mentioned in Material and methods of revised manuscript, to establish an animal model of liver fibrosis, mice were treated via intraperitoneal injections of CCl4 as our previously described (Journal of Cellular Biochemistry 2016;9999:1). Briefly, mice received CCl4 dissolved in mineral oil or mineral oil alone at a dose of 0.5 mL/kg body weight twice a week for 8 weeks to induce liver fibrosis. The control group received mineral oil alone at the same time. As suggested by the reviewer, we have included some sentences (see page 3, line 136-138 in revision).

Q-3 In their morphological and functional abnormalities in mitochondrial experiments, authors show a decrease of CIV-MTCO1 expression in the 8 weeks CCL4 mouse model of liver fibrogenesis. The blot showed in figure 1D does not show really a decrease of this expression. It seems to be very similar than the CII-SDHB complex which is not modify. Could authors explain this confused results? Could authors precise the number of animals and the method used to quantify the expression level in these experiments?

- As suggested by the reviewer, we have included the new representative results of OXPHOS complexes expression (see Figure 1E in revision).

Q-4. In their experiments to check the downregulation of the HtrA2/Omi expression in the CCL4 mouse model, authors show the decrease of the HtrA2/Omi expression during the 8 weeks of treatment. It appears to decrease after 4 weeks of treatment (figure 2B on the blot), could authors explain this latency time? The HtrA2/Omi expression was also evaluated in liver tissue from patient with healthy and fibrosis liver. Why the number n of sample analysis is different between this two conditions? Basically, for all of their experiments authors have to use an identical number n of samples / experiments in their groups.

- We have much experience in establishing a CCl4 -induced liver fibrosis mouse model along with previous studies (ACS Nano. 2010;4(6):3005 / Biomaterials. 2011;32(21):4951 / J Cell Biochem. 2017;118(8):2026 / Gut Liver. 2017;11(1):102). Based on these, we identified the histopathological features and changes in mitochondrial morphology during the progression of hepatic fibrosis were confirmed. Mice appear to be an early development of inflammation and fibrosis in the 4th week after starting the CCl4 -induced liver fibrosis and be more severe inflammation and fibrosis develop in the 8th week. In addition, the mitochondrial morphology observed by TEM were significantly impaired at 8 weeks compared to 4 weeks after CCl4 administration as shown below result. In Figure 2B, we showed that HtrA2/Omi expression gradually decreased with increasing liver fibrotic marker SMA expression and mitochondrial damage. As results, we chose the 8 weeks CCl4 treatment with severe liver fibrosis and severe mitochondrial damage. Furthermore, as suggested by the reviewer, in vitro and in vivo experiments could be performed using the same number of samples between all groups. However, considering the diversity of clinical samples and individual differences in animal model, the number of experimental groups was larger than that of the control group, and many other papers used the same method (Blood Adv. 2018; 2(5): 470 / Sci Rep. 2017;7(1):15532 /  BMC Complement Altern Med. 2014;14:449). For this reason, we conducted experiments by extracting liver tissue from various sites within one mouse subject. In addition, when liver fibrosis modeling is performed, the mouse suddenly dies of hepatotoxicity, and therefore, the number of tresses in the experimental group was more than in the normal experimental group.

Q-5. In their experiments to check the αSMA expression in the CCL4 mouse model, authors show an increase of this expression during the CCL4 treatment versus control. This protein is also up regulated during heart fibrosis after ischemia. What is the consequence of the increase of the αSMA expression on the liver fibrosis in regard of the heart fibrosis which is a similar phenomenon?

- In order to response the reviewer's comment, the pathogenesis of liver fibrosis would first be explained. As descripted in introduction section, liver fibrosis is a primary consequence of liver injury that results from chronic liver diseases, including chronic viral hepatitis, alcohol and metabolic liver disease. Persistent hepatocyte damage induces the release of cytokines and growth factors, such as transforming growth factor-β (TGF-β), which leads to the activation of hepatic stellate cells (HSCs). The transformation of quiescent HSCs into activated myofibroblasts results in the upregulation of α-SMA and the deposition of extracellular matrix, which play a pivotal role in the occurrence and development of liver fibrosis. Therefore, α-SMA expression is considered a marker of HSC activation, which, may reflect the degree of liver fibrosis.

Q-6. In their experiments in figure 4, authors show the consequences of the loss of the HtrA2/Omi in hepatocyte on mitochondria function and oxidative stress. They show that the loss of the HtrA2/Omi promotes an increase of the pro oxidant compounds as superoxide anion. What is the consequences of the loss of the HtrA2/Omi if the cells are treated with an anti-oxidant compound as the alpha tocopherol (vitamin e)?

- In previous study (PNAS 2008, 105(37):14106-11), they identified Mpv17l as a mitochondrial binding partner of HtrA2/Omi and a regulator of the mitochondrial function of the HtrA2/Omi to suppress mitochondrial ROS generation. They show that this antioxidant function of HtrA2/Omi is essential for stabilization of the inner mitochondrial membrane potential and mitochondrial retention of cytochrome c and HtrA2/Omi itself, thereby protecting cells against mitochondrial apoptosis signals, such as metabolic and cytokine stimuli. In addition, as our results show, the loss of HtrA2 / Omi breaks the antioxidant balance and as a result is more sensitive to oxidative stress. Reviewer's advice is very important in this regard. We will further study antioxidant mechanisms and liver fibrogenesis according to HtrA2 / Omi expression or protease activity.

Q-7. In their experiments in figure 7, authors show that rescue of HtrA2/Omi expression in CCL4-induced liver fibrosis mouse model protects mitochondrial damage of hepatocyte. The fibrosis is associated to the extracellular matrix production in the tissue with a scare fibrotic tissue formation. Could authors explain why there is not significant difference between mock and HtrA2/Omi in CCL4 injection group for expression level of mt-COX1/Ndufv1? Why authors did not do a collagen dosage in their different experimental conditions?

- For the preventive study, we showed that rescue experiment of HtrA2/Omi expression by hydrodynamic gene delivery protects mitochondrial damage of hepatocyte in CCl4-induced liver fibrosis mouse model. The injection of HtrA2/Omi plasmid DNA were given with CCl4 treatment twice weekly for 8 weeks. In our results as well as in many other papers (PNAS. 2012;109(24):9448 / World J Gastroenterol. 2019; 25(24): 3044), after cessation of the CCl4 toxic agent, mice spontaneously recovered for 1 month, and regression of liver fibrosis was evaluated by measuring collagen deposition and myofibroblast number. For these reasons, we injected HtrA2/Omi plasmid DNA together with CCl4 administration, and therefore the expression level of mt-COX1/Ndufv1 in livers treated with pFLAG-HtrA2/Omi was reduced, but not significant. However, as shown in Figure 6E, the hydroxyproline content was significantly lower in livers treated with pFLAG-HtrA2/Omi (1.1 μg/mg, P < 0.01) than in control CCl4-treated livers (2.17 μg/mg). Also, quantification indicated that the Sirius red-positive area was smaller (by 5.8%) in fibrotic livers from mice injected with pFLAG-HtrA2/Omi plasmids than in livers from CCl4-treated mice (Fig. 6A & 6D). These results suggest that HtrA2/Omi expression appears to reverse or at least prevent further progression of liver fibrosis.

Q-8. Authors used the Rompun and the Zoletil to anaesthetize their mice. Why did they use these anesthetic agents instead of classical agents such as xylazine / Ketamine?

- We commonly use Zoletil & Rompun or Xylazine & Ketamine to anesthetize mice in our country or our laboratory animals of research supporting Center for Medical Science of the Catholic University of Korea. Our laboratory animals are a fully accredited institution from AAALAC International and has been guided accordingly. As you know, both Rompun and Ketamine are known psychotropic Drugs. Ketamine is widely used for animal experiments, including humans, while Rompun is mainly used for animal experiments. Ketamin is detoxified by the liver and released by the kidneys, and its use in animals with renal failure or liver failure is limited (NDT Plus. 2008; 1(5): 310 / Urological Science 2017; 28(3):123 / Journal of Anesthesia & Critical Care 2019; 2(1):6). For these reasons, we have used Zoletil (xylazine) & Rompun for a long time in our Lab and it works very well without toxicity to liver or kidney and maintains deep anesthesia for perfusion.

Q-9 & Q-10.  Authors give some informations about data but these data were not shown (line 477). I think they have to show these datas in supplemental figure. In their experiments in figure 5, authors show the relative expression of hSMA and hColα1 mRNA. I am not sure to understand clearly the statistical analysis on the histograms and I am not sure of the significance between the different conditions. Could authors clarify this point?

- We apologize for any confusion this may have caused. As described in the previous study (Gut 2005;54:142), the human stellate cell line LX-2 cell was generated by spontaneous immortalization in low serum conditions. The phenotype of its cells is most like that of “activated” in vivo. Despite this activated phenotype, LX-2 lines can be quiesced by low serum condition. For these reasons, we performed the reversion of the activated LX-2 under serum free condition (S.F; preventive of inactivated HSC) and then examined the effects of HSC activation following treatment with mtDNA isolated from hepatocytes. In this process, already activated LX-2 cell seemed to have an error according to the sensitivity under experimental conditions such as serum-free and we were added the new results using the repeated experiments and the number of samples. Furthermore, we have included the relative intensity measurement of immunofluorescence for α-SMA in Fig.5B & 5D (see Figure 5A-5D; page 13, line 512-519 in revision).

Q-11. As mentioned before, some experiments have a different number of n, it will be better if this number is equal for each condition.

- As mentioned in Q-4, in vitro and in vivo experiments could be performed using the same number of samples between all groups. However, considering the diversity of clinical samples and individual differences in animal model, the number of experimental groups was larger than that of the control group, and many other papers used the same method (Blood Adv. 2018; 2(5): 470 / Sci Rep. 2017;7(1):15532 /  BMC Complement Altern Med. 2014;14:449). For this reason, we conducted experiments by extracting liver tissue from various sites within one mouse subject. In addition, when liver fibrosis modeling is performed, the mouse suddenly dies of hepatotoxicity, and therefore, the number of tresses in the experimental group was more than in the normal experimental group.

Q-12. After the list of references, the legend of the figure 1 is missing.

- We have added the legend of the Figure 1 according to the reviewer’s comment (see page 20, line 772-781  in revision).

Q-13. The figure 1 is not correct in the top left.

- We have edited the manuscript according to the reviewer’s comment (see page 8, Figure 1 in revision).

We hope that our revised manuscript has satisfactorily addressed the concerns of the Reviewers. We express our utmost appreciation and gratitude for your valuable remarks.

Yours sincerely, 

Round 2

Reviewer 1 Report

In the revised manuscript, the authors have clearly addressed all the reviewers' questions.

Author Response

Thank you for the excellent help. 

We have corrected the spelling and grammatical mistakes.

Reviewer 4 Report

Q-1. In their experiments on clinical samples, authors don’t give some precisions on the pharmacological treatments for the patients. Did authors make different groups in function of the pharmacological treatments? How many samples were used for the experiments? Do samples come from patients with the same liver pathology or not?

Æ As suggested by the reviewer, we have included some more detailed sentence about clinical samples (see page 2, line 117-121 in revision). We obtained five liver fibrosis tissues from patients with diagnosed chronic liver diseases who underwent liver transplantation (Seoul St. Mary’s Hospital, Seoul, South Korea) prior to 2010 and stored in liquid nitrogen. None of them had history of any treatment.

Ok with this answer from the authors

Q-2. In their study, authors establish an animal model of liver fibrosis by intraperitoneal injections of CCL4 during several weeks. This model was used for their different experiments versus a control. What is the control used in these experiments? A sham group with intraperitoneal injections of saline solution during the same period? Could authors clarify this point?

Æ Many studies reported that experimental liver fibrosis in mice can be induced by surgical intervention (bile duct ligation), genetic manipulation of fibrosis-related genes (Mdr2 knockout mice) or application of hepatotoxins (Fibrogenesis Tissue Repair 2013;6:19 / BMC Gastroenterol. 2010;10:79 / J. Clin. Invest. 1994;94:2481 / Current Pathobiology Reports 2014;2(4):143). In particular, the single or repeated administration of carbon tetrachloride (CCl4) has become one of the most commonly used experimental models for inducing toxin-mediated liver fibrosis. As mentioned in Material and methods of revised manuscript, to establish an animal model of liver fibrosis, mice were treated via intraperitoneal injections of CCl4 as our previously described (Journal of Cellular Biochemistry 2016;9999:1). Briefly, mice received CCl4 dissolved in mineral oil or mineral oil alone at a dose of 0.5 mL/kg body weight twice a week for 8 weeks to induce liver fibrosis. The control group received mineral oil alone at the same time. As suggested by the reviewer, we have included some sentences (see page 3, line 136-138 in revision).

Ok with this answer from the authors

Q-3 In their morphological and functional abnormalities in mitochondrial experiments, authors show a decrease of CIV-MTCO1 expression in the 8 weeks CCL4 mouse model of liver fibrogenesis. The blot showed in figure 1D does not show really a decrease of this expression. It seems to be very similar than the CII-SDHB complex which is not modify. Could authors explain this confused results? Could authors precise the number of animals and the method used to quantify the expression level in these experiments?

Æ As suggested by the reviewer, we have included the new representative results of OXPHOS complexes expression (see Figure 1E in revision).

Ok with this answer from the authors

Q-4. In their experiments to check the downregulation of the HtrA2/Omi expression in the CCL4 mouse model, authors show the decrease of the HtrA2/Omi expression during the 8 weeks of treatment. It appears to decrease after 4 weeks of treatment (figure 2B on the blot), could authors explain this latency time? The HtrA2/Omi expression was also evaluated in liver tissue from patient with healthy and fibrosis liver. Why the number n of sample analysis is different between this two conditions? Basically, for all of their experiments authors have to use an identical number n of samples / experiments in their groups.

Æ We have much experience in establishing a CCl4 -induced liver fibrosis mouse model along with previous studies (ACS Nano. 2010;4(6):3005 / Biomaterials. 2011;32(21):4951 / J Cell Biochem. 2017;118(8):2026 / Gut Liver. 2017;11(1):102). Based on these, we identified the histopathological features and changes in mitochondrial morphology during the progression of hepatic fibrosis were confirmed. Mice appear to be an early development of inflammation and fibrosis in the 4th week after starting the CCl4 -induced liver fibrosis and be more severe inflammation and fibrosis develop in the 8th week. In addition, the mitochondrial morphology observed by TEM were significantly impaired at 8 weeks compared to 4 weeks after CCl4 administration as shown below result.

In Figure 2B, we showed that HtrA2/Omi expression gradually decreased with increasing liver fibrotic marker SMA expression and mitochondrial damage. As results, we chose the 8 weeks CCl4 treatment with severe liver fibrosis and severe mitochondrial damage. Furthermore, as suggested by the reviewer, in vitro and in vivo experiments could be performed using the same number of samples between all groups. However, considering the diversity of clinical samples and individual differences in animal model, the number of experimental groups was larger than that of the control group, and many other papers used the same method (Blood Adv. 2018; 2(5): 470 / Sci Rep. 2017;7(1):15532 /  BMC Complement Altern Med. 2014;14:449). For this reason, we conducted experiments by extracting liver tissue from various sites within one mouse subject. In addition, when liver fibrosis modeling is performed, the mouse suddenly dies of hepatotoxicity, and therefore, the number of tresses in the experimental group was more than in the normal experimental group.

I understand the answer from the authors but in my opinion results are statistically stronger with an equal group’s n number and there is always experimental procedures to have it !

Q-5. In their experiments to check the αSMA expression in the CCL4 mouse model, authors show an increase of this expression during the CCL4 treatment versus control. This protein is also up regulated during heart fibrosis after ischemia. What is the consequence of the increase of the αSMA expression on the liver fibrosis in regard of the heart fibrosis which is a similar phenomenon?

Æ In order to response the reviewer's comment, the pathogenesis of liver fibrosis would first be explained. As descripted in introduction section, liver fibrosis is a primary consequence of liver injury that results from chronic liver diseases, including chronic viral hepatitis, alcohol and metabolic liver disease. Persistent hepatocyte damage induces the release of cytokines and growth factors, such as transforming growth factor-β (TGF-β), which leads to the activation of hepatic stellate cells (HSCs). The transformation of quiescent HSCs into activated myofibroblasts results in the upregulation of α-SMA and the deposition of extracellular matrix, which play a pivotal role in the occurrence and development of liver fibrosis. Therefore, α-SMA expression is considered a marker of HSC activation, which, may reflect the degree of liver fibrosis.

Ok with this answer from the authors

Q-6. In their experiments in figure 4, authors show the consequences of the loss of the HtrA2/Omi in hepatocyte on mitochondria function and oxidative stress. They show that the loss of the HtrA2/Omi promotes an increase of the pro oxidant compounds as superoxide anion. What is the consequences of the loss of the HtrA2/Omi if the cells are treated with an anti-oxidant compound as the alpha tocopherol (vitamin e)?

Æ In previous study (PNAS 2008, 105(37):14106-11), they identified Mpv17l as a mitochondrial binding partner of HtrA2/Omi and a regulator of the mitochondrial function of the HtrA2/Omi to suppress mitochondrial ROS generation. They show that this antioxidant function of HtrA2/Omi is essential for stabilization of the inner mitochondrial membrane potential and mitochondrial retention of cytochrome c and HtrA2/Omi itself, thereby protecting cells against mitochondrial apoptosis signals, such as metabolic and cytokine stimuli. In addition, as our results show, the loss of HtrA2 / Omi breaks the antioxidant balance and as a result is more sensitive to oxidative stress. Reviewer's advice is very important in this regard. We will further study antioxidant mechanisms and liver fibrogenesis according to HtrA2 / Omi expression or protease activity.

Ok with this answer from the authors

Q-7. In their experiments in figure 7, authors show that rescue of HtrA2/Omi expression in CCL4-induced liver fibrosis mouse model protects mitochondrial damage of hepatocyte. The fibrosis is associated to the extracellular matrix production in the tissue with a scare fibrotic tissue formation. Could authors explain why there is not significant difference between mock and HtrA2/Omi in CCL4 injection group for expression level of mt-COX1/Ndufv1? Why authors did not do a collagen dosage in their different experimental conditions?

Æ For the preventive study, we showed that rescue experiment of HtrA2/Omi expression by hydrodynamic gene delivery protects mitochondrial damage of hepatocyte in CCl4-induced liver fibrosis mouse model. The injection of HtrA2/Omi plasmid DNA were given with CCl4 treatment twice weekly for 8 weeks. In our results as well as in many other papers (PNAS. 2012;109(24):9448 / World J Gastroenterol. 2019; 25(24): 3044), after cessation of the CCl4 toxic agent, mice spontaneously recovered for 1 month, and regression of liver fibrosis was evaluated by measuring collagen deposition and myofibroblast number. For these reasons, we injected HtrA2/Omi plasmid DNA together with CCl4 administration, and therefore the expression level of mt-COX1/Ndufv1 in livers treated with pFLAG-HtrA2/Omi was reduced, but not significant. However, as shown in Figure 6E, the hydroxyproline content was significantly lower in livers treated with pFLAG-HtrA2/Omi (1.1 μg/mg, P < 0.01) than in control CCl4-treated livers (2.17 μg/mg). Also, quantification indicated that the Sirius red-positive area was smaller (by 5.8%) in fibrotic livers from mice injected with pFLAG-HtrA2/Omi plasmids than in livers from CCl4-treated mice (Fig. 6A & 6D). These results suggest that HtrA2/Omi expression appears to reverse or at least prevent further progression of liver fibrosis.

Ok with this answer from the authors

Q-8. Authors used the Rompun and the Zoletil to anaesthetize their mice. Why did they use these anesthetic agents instead of classical agents such as xylazine / Ketamine?

Æ We commonly use Zoletil & Rompun or Xylazine & Ketamine to anesthetize mice in our country or our laboratory animals of research supporting Center for Medical Science of the Catholic University of Korea. Our laboratory animals are a fully accredited institution from AAALAC International and has been guided accordingly. As you know, both Rompun and Ketamine are known psychotropic Drugs. Ketamine is widely used for animal experiments, including humans, while Rompun is mainly used for animal experiments. Ketamin is detoxified by the liver and released by the kidneys, and its use in animals with renal failure or liver failure is limited (NDT Plus. 2008; 1(5): 310 / Urological Science 2017; 28(3):123 / Journal of Anesthesia & Critical Care 2019; 2(1):6). For these reasons, we have used Zoletil (xylazine) & Rompun for a long time in our Lab and it works very well without toxicity to liver or kidney and maintains deep anesthesia for perfusion.

Ok with this answer from the authors

Q-9 & Q-10.  Authors give some informations about data but these data were not shown (line 477). I think they have to show these datas in supplemental figure. In their experiments in figure 5, authors show the relative expression of hSMA and hColα1 mRNA. I am not sure to understand clearly the statistical analysis on the histograms and I am not sure of the significance between the different conditions. Could authors clarify this point?

Æ We apologize for any confusion this may have caused. As described in the previous study (Gut 2005;54:142), the human stellate cell line LX-2 cell was generated by spontaneous immortalization in low serum conditions. The phenotype of its cells is most like that of “activated” in vivo. Despite this activated phenotype, LX-2 lines can be quiesced by low serum condition. For these reasons, we performed the reversion of the activated LX-2 under serum free condition (S.F; preventive of inactivated HSC) and then examined the effects of HSC activation following treatment with mtDNA isolated from hepatocytes. In this process, already activated LX-2 cell seemed to have an error according to the sensitivity under experimental conditions such as serum-free and we were added the new results using the repeated experiments and the number of samples. Furthermore, we have included the relative intensity measurement of immunofluorescence for α-SMA in Fig.5B & 5D (see Figure 5A-5D; page 13, line 512-519 in revision).

Ok with this answer from the authors

Q-11. As mentioned before, some experiments have a different number of n, it will be better if this number is equal for each condition.

Æ As mentioned in Q-4, in vitro and in vivo experiments could be performed using the same number of samples between all groups. However, considering the diversity of clinical samples and individual differences in animal model, the number of experimental groups was larger than that of the control group, and many other papers used the same method (Blood Adv. 2018; 2(5): 470 / Sci Rep. 2017;7(1):15532 /  BMC Complement Altern Med. 2014;14:449). For this reason, we conducted experiments by extracting liver tissue from various sites within one mouse subject. In addition, when liver fibrosis modeling is performed, the mouse suddenly dies of hepatotoxicity, and therefore, the number of tresses in the experimental group was more than in the normal experimental group.

I understand the answer from the authors but in my opinion results are statistically stronger with an equal group’s n number and there is always experimental procedures to have it !

Q-12. After the list of references, the legend of the figure 1 is missing.

 Æ We have added the legend of the Figure 1 according to the reviewer’s comment (see page 20, line 772-781  in revision).

Ok with this answer from the authors

Q-13. The figure 1 is not correct in the top left.

Æ We have edited the manuscript according to the reviewer’s comment (see page 8, Figure 1 in revision).

Ok with this answer from the authors

Author Response

Thank you for giving me advice about an equal group’s n number.

Reviewer's comment is very important in this regard.

We will consider your advice from the next study.